# Functional effects of distinct innervation styles of pyramidal cells by fast spiking cortical interneurons

**Yoshiyuki Kubota[1,2,3]\*, Satoru Kondo[3,4], Masaki Nomura[3,5†], Sayuri Hatada[1], Noboru Yamaguchi[1], Alsayed A Mohamed[1,6], Fuyuki Karube[1,3,9], Joachim Lübke[7,8,10], Yasuo Kawaguchi[1,2,3]**

[1]Division of Cerebral Circuitry, National Institute for Physiological Sciences, Okazaki, Japan; [2]Department of Physiological Sciences, The Graduate University for Advanced Studies, Okazaki, Japan; [3]Core Research for Evolutional Science and Technology, Japan Science and Technology Agency, Tokyo, Japan; [4]Department of Molecular Physiology, Kyushu University, Fukuoka, Japan; [5]Department of Mathematics, Kyoto University, Kyoto, Japan; [6]Department of Anatomy and Embryology, South Valley University, Qena, Egypt; [7]Institute of Neuroscience and Medicine, Research Centre Jülich, Jülich, Germany; [8]Department of Psychiatry, Psychotherapy and Psychosomatics, Medical Faculty, RWTH/University Hospital Aachen, Aachen, Germany; [9]Laboratory of Neural Circuitry, Graduate School of Brain Science, Doshisha University, Kyotanabe, Japan; [10]JARA Translational Brain Medicine, Jülich/Aachen, Germany

\*For correspondence: yoshiy@nips.ac.jp

**Present address:** †Center for iPS Cell Research and Application, Kyoto University, Kyoto, Japan

**Competing interests:** The authors declare that no competing interests exist.

**Abstract** Inhibitory interneurons target precise membrane regions on pyramidal cells, but differences in their functional effects on somata, dendrites and spines remain unclear. We analyzed inhibitory synaptic events induced by cortical, fast-spiking (FS) basket cells which innervate dendritic shafts and spines as well as pyramidal cell somata. Serial electron micrograph (EMg) reconstructions showed that somatic synapses were larger than dendritic contacts. Simulations with precise anatomical and physiological data reveal functional differences between different innervation styles. FS cell soma-targeting synapses initiate a strong, global inhibition, those on shafts inhibit more restricted dendritic zones, while synapses on spines may mediate a strictly local veto. Thus, FS cell synapses of different sizes and sites provide functionally diverse forms of pyramidal cell inhibition.

## Introduction

Microcircuits of cerebral cortex are composed of excitatory pyramidal cells and different types of GABAergic interneurons. Inhibitory circuits regulate cortical activity (*Kubota et al., 2011b*; *Lee et al., 2012*; *Kubota, 2014*), development and plasticity (*Hensch, 2005*; *Donato et al., 2013*). Perturbed inhibitory function is associated with pathologies including epilepsy, autism and schizophrenia (*Rubenstein and Merzenich, 2003*; *Gonzalez-Burgos et al., 2010*). However, mechanisms controlling inhibitory synaptic actions are incompletely understood. For instance, inhibitory synapses target multiple membrane domains of pyramidal cells: soma, axon initial segment, dendritic shafts and spines (*Kisvarday et al., 1985*; *Kawaguchi and Kubota, 1998*; *Szabadics et al., 2006*; *Kubota et al., 2007*; *Jiang et al., 2013*). Contacts at these different sites produce inhibitory postsynaptic potentials (IPSP) with different properties (*Miles et al., 1996*; *Xue et al., 2014*).

Recent data suggests IPSCs generated by FS basket cells may be matched to the level of synaptic excitation in cortical pyramidal cells (*Xue et al., 2014*), and differ with target cell subtypes

**eLife digest** The brain contains millions of cells called neurons that communicate with one another as part of complex circuits. To send information around these circuits, neurons 'fire' electrical signals along their length. These trigger the release of chemicals across a structure—known as the synapse—that forms a connection with a neighboring cell. Different types of neurons affect their neighbors in different ways. For example, signals from a pyramidal cell make it more likely that the next cell in the circuit will fire, whereas a signal sent by an inhibitory interneuron has the opposite effect. Pyramidal cells and interneurons make up the circuits in the brain's outer layer—the cortex.

Despite their opposing roles, these cells share the same basic structure. Each consists of a cable-like axon that can efficiently transmit electrical signals, and a cell body that contains the nucleus. The cell body bears numerous short branches called dendrites, which are in turn covered in bump-like protrusions called spines. Synapses typically form between the end of one cell's axon and a dendrite on another cell. However, synapses can also form between the end of an axon and an individual dendritic spine, or the end of an axon and a cell body.

Models of inhibitory synapses—connections from interneurons that inhibit pyramidal cells—tend to assume that these three types of connection are equivalent. However, Kubota et al. have now combined electron microscopy with electrode recordings of the activity of pairs of connected cells to show that the size and ability of inhibitory synapses to inhibit signaling varies depending on their location. Specifically, inhibitory synapses that form with the cell bodies of pyramidal cells are larger and inhibit signaling more strongly than those that form with dendrites, which are in turn larger and more inhibitory than those on dendritic spines.

Thus, depending on the point at which an interneuron contacts a pyramidal cell, it can inhibit signaling throughout the entire cell body, or only across a dendrite, or even just within a single dendritic spine. Incorporating this information into computer models of the brain will improve how accurately they simulate how the brain works. It will also help when modeling disorders in which inhibitory networks are disrupted, such as schizophrenia and depression.

(*Lee et al., 2014*). Unitary inhibitory postsynaptic currents (uIPSCs) are significantly smaller in neurons of Disc1 mice, a genetic model of depression, and may underlie reduced low-gamma oscillations in the frontal cortex (*Sauer et al., 2015*). GABA receptors on spine heads are thought to control local synaptic excitation (*Chiu et al., 2013*). However the structural basis for these effects remains unclear. Modeling studies assume that somatic, dendritic shaft and spine inhibition is mediated by pre-synaptic elements of identical size and strength (*Gidon and Segev, 2012*). In contrast, excitatory synaptic terminals vary in size and their strength is correlated with terminal size (*Holderith et al., 2012*). We therefore examined this point for cortical inhibition by correlating structural and functional properties of synapses of FS basket cells on layer V (L5) pyramidal cells of rat frontal cortex. Physiological and anatomical data from paired recordings let us simulate the dendro-somatic conduction of the effects of inhibitory synapses made on different membrane sites on pyramidal cells. We show that synapses made by FS basket cells on the soma and on dendritic shafts and spines have dramatically different functional effects.

## Results

### Double recording

Crossed-corticostriatal (CCS) 'slender untufted' pyramidal cells (*Larkman and Mason, 1990 Morishima and Kawaguchi, 2006*) are a discrete neuronal population in L5. We investigated connections between FS basket cells and CCS pyramidal cells, identified by injecting a fluorescent retrograde tracer into the contralateral striatum (*Figure 1—figure supplement 1*). IPSCs were evoked in postsynaptic CCS pyramidal cell soma by single APs in FS basket cells (*Figure 1—figure supplement 2*). With pyramidal cell membrane potential maintained at −65 mV, IPSCs reversed on average at −52.5 mV (*Figure 1—figure supplement 2C*), providing a mean driving force of 12.5 mV. After recording and biocytin-filling, axonal and dendritic morphology and the number and distribution of possible synaptic contacts from each coupled pair were analyzed (n = 10) using *Neurolucida* software (*Figure 1B–E,G–I*, *Figure 2A–D,F–I*). Paired recordings were made from neighboring cells (*Table 1*, inter-somatic

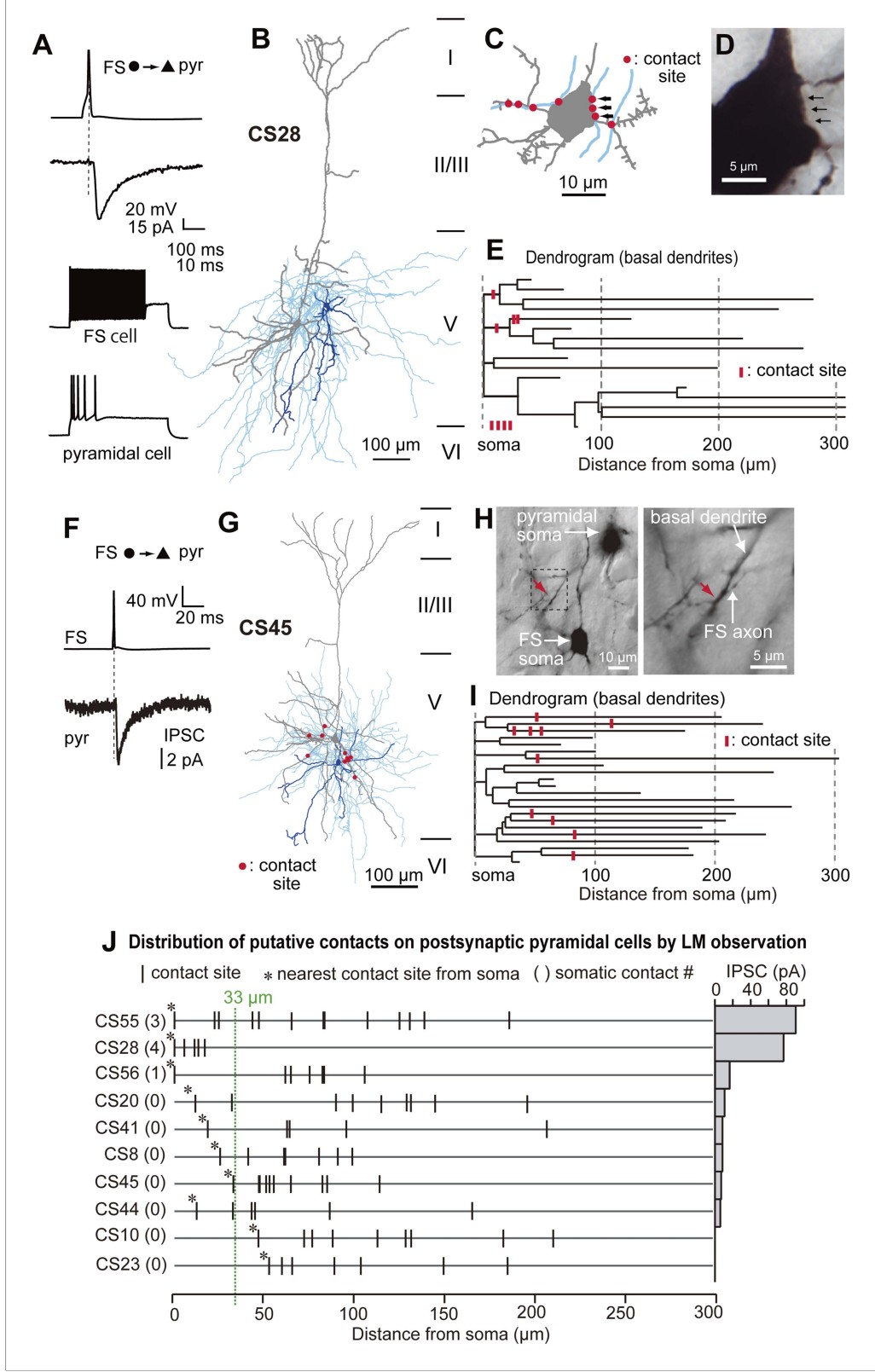

**Figure 1**. Paired recording between FS basket cells and CCS pyramidal cells in L5. (**A–E**) Structural and functional characteristics of pair CS28. (**A**) The presynaptic FS basket cell shows a fast-spiking (upper left) and the postsynaptic pyramidal cell displayed a regular spiking behavior (bottom left). Average IPSC response in the pyramidal cell

*Figure 1. continued on next page*

*Figure 1. Continued*

(bottom right) to a single action potential (AP) elicited in the FS basket cell (upper right). (**B**) Reconstruction of the neuron pair. The somatodendritic domain of the presynaptic FS basket cell is shown in blue, the axonal arborization in sky blue, and the somatodendritic domain of the postsynaptic pyramidal cell in gray. (**C**) Illustration showing the number and distribution of putative synaptic contacts (red dots) established by the FS basket cell axonal collaterals on the soma and proximal dendritic segments of the postsynaptic pyramidal cell. (**D**) LMg of the pyramidal cell soma with its inhibitory synaptic contacts (arrows) illustrated in (**C**). (**E**) Dendrogram of the pyramidal cell basal dendrites with putative contact sites (red bars). (**F–I**) Structural and functional characteristics of pair CS45. (**F**) Averaged IPSC in the pyramidal cell in response to a single AP in the presynaptic FS basket cell. (**G**) Reconstruction of the cell pair. Same color code as in (**B**) with putative synaptic contacts (red). Note that synaptic contacts were exclusively found on dendrites. (**H**) Low power LMg of the cell pair showing a putative contact site on the basal dendrite of the pyramidal cell (red arrow) by the FS basket cell axon at low (left panel) and high (right panel) magnification. (**I**) Dendrogram of the basal dendrites of the pyramidal cell with ten LM-identified contact sites (red bars). (**J**) Summary diagram showing the number and distribution of putative contacts established on postsynaptic pyramidal cell somata and dendrites for all investigated pairs. The corresponding averaged IPSC peak amplitude is shown on the right. For the last two pairs, no IPSCs were detectable despite the presence of LM-identified contact sites.

The following figure supplements are available for figure 1:

**Figure supplement 1**. The CCS pyramidal cell in layer V identified by retrograde fluorescent tracer.

**Figure supplement 2**. Physiological properties of IPSCs evoked in CCS pyramidal cells in L5.

distance: 44.5 ± 23.7 μm, 20.6–66.6 μm, n = 10). There was typically a large overlap of the basal dendrites of postsynaptic pyramidal cells and the axonal arbor of presynaptic FS basket cells (***Figure 1B, G***, ***Figure 2B,G***, ***Figure 2—figure supplement 1***). In three cell pairs, FS basket cell axons established putative synaptic contacts on the soma and dendrites of a postsynaptic CCS pyramidal cell (***Figure 1J***, upper three lines). In seven pairs, synaptic contacts were located exclusively on dendrites at various distances from the soma (***Figure 1J***, lower 7 lines). The number of putative synaptic contacts was 5–14 (8.2 ± 4.8, 10 pairs). Most light microscopic contacts were made where FS basket cell axons crossed basal pyramidal cell dendrites (***Figure 1D,E,H,I***, ***Figure 2D,I***, ***Figure 3B***) (***Marlin and Carter, 2014***). The distance from the soma to dendritic contacts was 5.8–208.4 μm with a mean value of 82.5 ± 50.0 μm. Peak IPSC amplitude was larger in pairs with putative somatic contacts than those when contacts were exclusively dendritic (***Figure 1J***). Transmission never failed for pairs with somatic contacts but failures occurred with dendritic contacts (***Table 1***). Mean IPSC amplitude, from pairs with only dendritic contacts, was reduced at increasing distances from the soma to the nearest contact (***Figure 1J***). IPSCs were not detected in two pairs, where light microscopy (LM) suggested 7 and 9 contacts were made at distances further than 33 μm from the soma (***Figure 1J***, lower 2 lines). In each case the pyramidal cell elicited large EPSC in the interneurons (***Table 1***).

We found large differences in IPSC amplitude evoked by FS cells in L5 pyramidal cells (***Figure 1A, F,J***, ***Figure 2E,J***). Large IPSCs were found in two pairs with somatic synaptic contacts. The size of IPSCs in the other pair with somatic/dendritic contacts was smaller (***Figure 1J***). Higher numbers of putative somatic terminals were correlated with larger synaptic events (***Figure 2C,D,H,I***). Thus the number of intersections of the presynaptic FS cell axon fibers within 18 μm from somatic center were larger in the pair CS55 with an IPSC of amplitude −91.3 pA than in pair CS56 where IPSC amplitude was −17.3 pA (***Figure 2—figure supplement 2***).

## Synapses identified by 3D reconstructions from serial EMgs

The number of synaptic terminals was verified and their size was measured using electron microscopy (EM). Junctional size governs transmitter release probability (***Holderith et al., 2012***) and docking sites (***Pulido et al., 2015***), with the number of postsynaptic receptors (***Nusser et al., 1997***; ***Tanaka et al., 2005***) which determines synaptic current amplitude. All putative synaptic contacts (***Figure 2D,I***) were completely reconstructed from serial EMgs (***Figure 3***, ***Figure 3—figure supplement 1***) for measurement of synaptic junction and dendritic cross sectional areas. Similar data from sixty one dendritic

segments (mean length 16.8 ± 6.8 µm) of the CS56 postsynaptic pyramidal cell and the entire soma of the pyramidal cell (*Figure 4*) was also used in neuron simulations (*Kubota et al., 2011a*).

EM analysis let us verify possible synaptic contacts from LM. For the pair CS56, 3 of 7 possible contacts were verified by EM, but no synaptic contact was made at 4 other potential sites (*Figure 4*). One putative LM contact was resolved as three distinct *en passant* boutons (S1–S3 in *Figure 5A–E*) and another somatic contact was detected only by EM (S4, *Figure 5—figure supplement 1*). The other two verified contacts terminated on spine heads (Sp2, Sp3 in *Figure 6A,C*). One with a thin dendrite (D1 in *Figure 5F,G,I*, *Figure 6A,C*) and nearby spine head (Sp1, *Figure 5F–I*, *Figure 6A,C*) were detected only by EM. The junctional area of synapses made by single interneurons varied strikingly with the post-synaptic site that is innervated. For somatic synapses junctional area was 0.194–0.350 µm², it was 0.102 µm² for synapses with dendritic shafts and 0.042–0.056 µm² for synapses onto spine heads (*Figure 6F*, *Table 2*). Axonal bouton volume was linearly correlated with synaptic junction area (*Figure 6—figure supplement 1A*).

Fourteen potential contacts, 3 at somatic and 11 at dendritic sites, were identified by LM for the pair CS55 (*Figure 2I*). Complete EM reconstruction of the post-synaptic soma let us explore sites obscured in LM where axon crossed the soma (*Figure 5J–N*, *Figure 5—figure supplement 2*) and revealed 13 synaptic contacts (S1–S13, *Figure 5K-N*, *Figure 6B,E,G*). Eight terminals made onto dendrites and spine heads less than 33 µm away from the soma presumably contributed to the somatic IPSC (*Figure 5J*, *Figure 6B,D,E,G*). Three dendritic shaft synapses (D5–D7), were located further than 33 µm from soma. Two potential LM contacts showed 2 synaptic contact sites, each. Four potential LM contacts were discounted from EM data (*Figure 5—figure supplement 3*), 2 potential LM contacts were not analyzed by EM (*Figure 6E*), and 4 synapses were only evident in EM. 3D EM reconstructions of all synapses (CS55 and CS56) showed that synaptic area was larger for somatic than dendritic contacts (*Figure 6F,G*, *Table 3*) and decreased continuously with distance from the soma.

Numbers of synaptic contacts were defined for two further neuron pairs, CS44 and CS23, by serial EMgs (*Figure 1J*). In the CS44 cell pair the closest confirmed synaptic contact was 32 µm distant from the soma, consistent with the inverse relation between synapse distance from the soma and the peak IPSC amplitude (*Figure 6E*). In pair CS23, EM verified five dendritic synaptic contacts with the nearest contact site 53 µm from the soma. Physiological analysis revealed the connection was nearly silent (*Figure 6E*). IPSCs induced by single FS interneurons at dendritic shaft synapses at 32 µm from soma (CS44) were detected with a somatic electrode, but with our recording configuration, IPSCs generated by terminals at 47 µm (CS10) and 53 µm (CS23) from the soma were not detected.

Three types of FS basket cell innervation can then be distinguished. Multiple synapses made with the soma or proximal dendrites of L5 CCS pyramidal cell produce large IPSCs, weaker somatic and proximal dendritic innervation produce intermediate IPSCs, while IPSCs are small or absent when synapses terminate exclusively on dendrites. From all paired records, 28.4 ± 7.6% (17.2–43.1%) of FS interneuron terminals contacted cell somata (*Figure 5*, *Figure 5—figure supplement 3A*, *Table 4*), consistent with previous data (*Karube et al., 2004*). We note that an FS cell that innervates only dendrites of one L5 pyramidal cell, may contact somatic sites of other postsynaptic neurons (*Figure 7*).

## Simulation analysis of IPSC conduction

Excitatory synaptic currents are correlated with synaptic size (*Holderith et al., 2012*). At larger synaptic junctions, $Ca^{2+}$ entry into presynaptic terminals is greater, transmitter release probability is increased (*Holderith et al., 2012*) and the number of postsynaptic receptors is larger (*Nusser et al., 1997*). We tested this relation for inhibitory transmission by comparing summed synaptic junction area with maximal IPSC amplitude for pairs CS56 and CS55. Maximal IPSCs (*Table 5*) were assumed to occur when all somatic and proximal dendritic terminals (<33 µm) (*Figure 5A–E*) released transmitter. The unit electrical charge was calculated as the maximum charge divided by the summed junction area of S1–S4: 326.1 fC/0.95 µm², or, 343.3 fC µm⁻² for pair CS56, and S1–S13, D1–D4, Sp1–Sp4: 1057.8 fC/3.011 µm², or 351.3 fC µm⁻² for pair CS55 (*Table 6*). This parameter was similar for the two connections, suggesting that currents are well correlated with synaptic junction area. Thus at these inhibitory synapses, conductance can be calculated from junctional area based on the unit IPSC electric charge using morphologically realistic CS56 postsynaptic pyramidal model cell based on our measurement of the cell dimensions (see 'Materials and method', *Table 2*).

Inhibitory synaptic connections made by FS basket cell axons terminate on the soma, dendritic shafts or spines of L5 CCS pyramidal cells (*Kubota et al., 2007*). We asked how these differences in

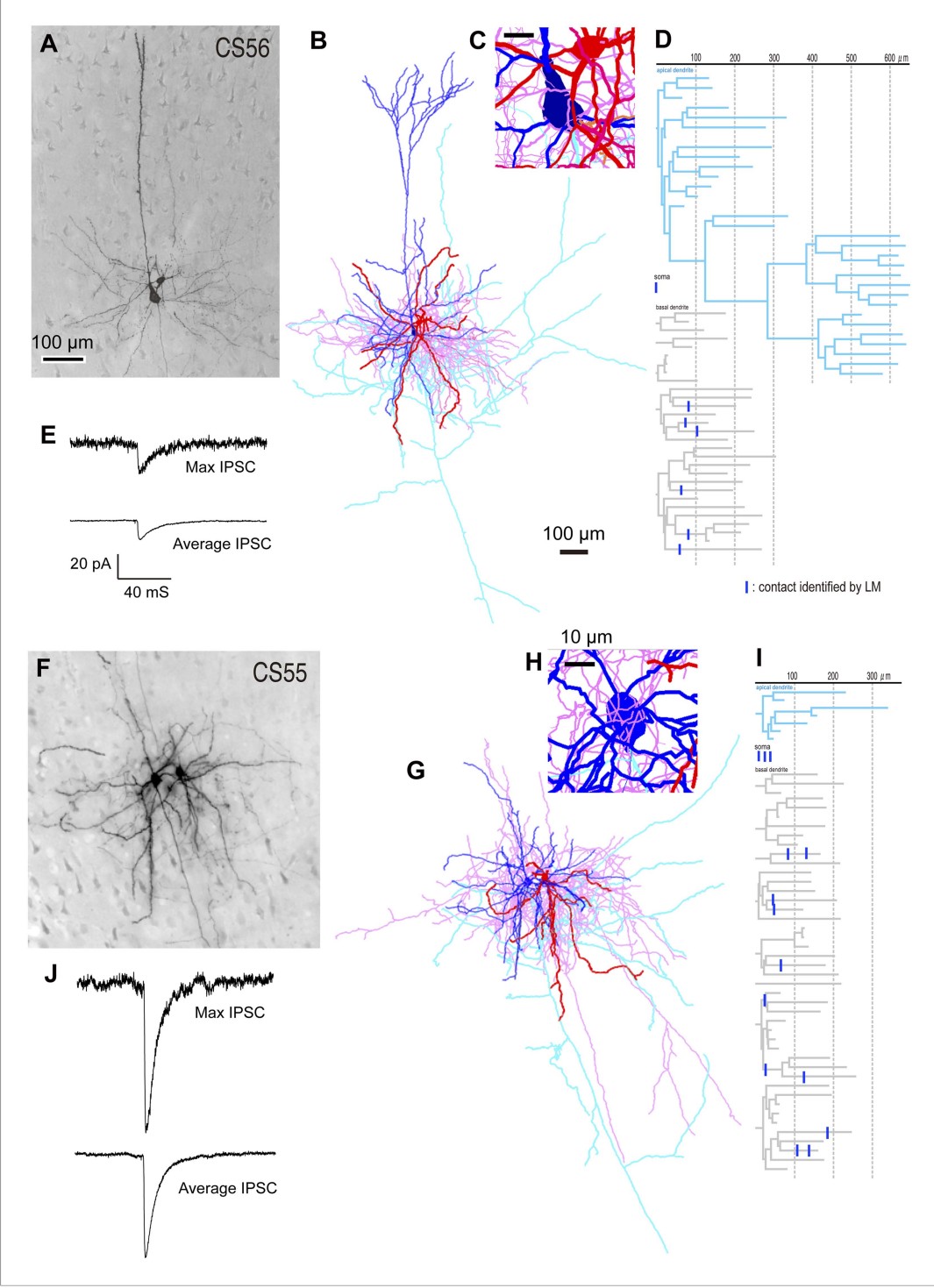

**Figure 2**. Different unitary IPSCs induced by single FS basket cells in L5 CCS pyramidal cells. (**A**) Pre-synaptic basket cell and post-synaptic pyramidal cell. Light micrograph (LMg) of the CS56 pair. (**B**) Reconstruction of pyramidal cell soma-dendrites (blue) and axon (sky blue), basket cell soma-dendrites (red) and axon (pink). (**C**) Close-up of the pyramidal cell soma. Scale, 10 µm. (**D**) Putative synaptic contacts (blue bars) shown on dendrogram including basal (gray) and apical (sky blue) dendrites. (**E**) Maximum (upper) and averaged (lower) IPSCs evoked by single FS basket cell spikes. (**F**) Pre-synaptic basket cell and post-synaptic pyramidal cell. LMg of CS55 pair. Scale is as in A. (**G**) Reconstruction. Scale is as in B. (**H**) Close-up of the pyramidal cell soma. (**I**) Dendrogram with putative synaptic contact sites (blue bar). (**J**) Maximum (upper) and average (lower) IPSCs evoked by single FS basket cell APs. Scale is as in E.

*Figure 2. continued on next page*

*Figure 2. Continued*

The following figure supplements are available for figure 2:

**Figure supplement 1**. Drawings of the paired recording between FS basket cells and CCS pyramidal cells in L5.
**Figure supplement 2**. Sholl analysis of presynaptic FS basket cell axon to postsynaptic CCS pyramidal cell soma center.

synaptic site and junctional size affect function in simulations based on our measurements of synaptic currents and dimensions. IPSC propagation was examined on an electrotonic simulation of the pyramidal cell from pair CS56. Injecting a 0.11 nS current on the spine head of Sp1 (*Table 2*) resulted in a strong 0.78 mV hyperpolarization of the spine, but only 0.12 mV was transmitted to the basal dendrite and 0.07 mV to the soma (*Figure 8A,C,K*). The peak synaptic current was 1.27 pA at the spine head, and 0.81 pA at the soma (*Figure 8B*). At noise levels of ∼10 pA (*Figure 1—figure supplement 2B*), a spine-head IPSC would not be detected at the soma. The spine neck effectively isolated the spine head from the dendritic shaft (neck length, 0.5 μm; diameter, 0.07 μm; volume, 0.043 μm$^3$; resistance, 500 MΩ [*Harnett et al., 2012*]). Thus spine inhibition did not change nearby dendritic shaft or somatic potential (*Araya et al., 2006*). In contrast, injecting a 0.21 nS synaptic current on the dendritic shaft (D1) (*Table 2*) caused a hyperpolarization of 0.23 mV on the shaft and

**Table 1**. Synapse properties of pair recordings

| | Amplitude (pA) | | | | | |
|---|---|---|---|---|---|---|
| | mean | sd | max | Success rate | Neurolucida analysis | Distance from soma (μm) |
| IPSC | | | | | | |
| CS4 | −5.7 | 5.1 | −19.5 | 0.6 | | |
| CS8 | −8.6 | 4.0 | −17.6 | 0.5 | yes | 48.8 |
| CS20 | −10.9 | 5.3 | −27.4 | 0.9 | yes | 51.8 |
| CS21* | −7.6 | 3.2 | −17.8 | 0.6 | | |
| CS22* | −8.0 | 3.4 | −14.5 | 0.6 | | |
| CS28 | −76.9 | 20.9 | −107.3 | 1.0 | yes | 48.8 |
| CS36* | −6.5 | 2.6 | −12.5 | 0.4 | | |
| CS41 | −8.6 | 4.3 | −20.8 | 0.7 | yes | 41.3 |
| CS44 | −6.2 | 2.1 | −12.6 | 0.5 | yes | 66.6 |
| CS45* | −7.1 | 4.1 | −21.2 | 0.7 | yes | 53.2 |
| CS55 | −91.3 | 11.2 | −111.0 | 1.0 | yes | 35.8 |
| CS56 | −17.3 | 3.0 | −24.9 | 1.0 | yes | 20.6 |
| CS61 | −9.6 | 4.6 | −22.2 | 0.8 | | |
| CS62 | −36.4 | 14.0 | −69.5 | 1.0 | | |
| EPSC | | | | | | |
| CS10 | 67.5 | 22.2 | 109.3 | 1.0 | yes | 26.5 |
| CS21* | 18.6 | 8.7 | 44.5 | 0.9 | | |
| CS22* | 70.9 | 38.3 | 201.6 | 1.0 | | |
| CS23 | 45.3 | 14.2 | 83.4 | 1.0 | yes | 51.2 |
| CS36* | 4.4 | 1.0 | 6.5 | 0.5 | | |
| CS45* | 43.1 | 19.0 | 86.1 | 1.0 | yes | 53.2 |

*Reciprocal connection between FS and pyramidal cell was observed.

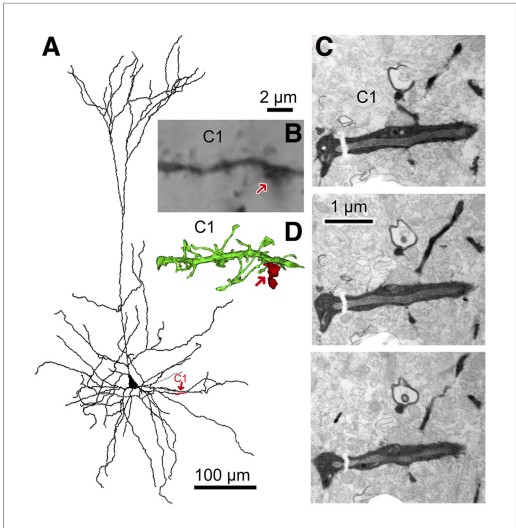

**Figure 3**. 3D reconstruction from serial EMgs. (**A**) *Neurolucida* reconstruction of the postsynaptic pyramidal cell of the CS56 pair. A dendritic segment (C1) is given in red and marked by red arrow. (**B**) Corresponding LMg of the dendritic segment C1 (focus stack image). The FS basket cell axon terminal is indicated by arrow. (**C**) EMgs from three adjacent ultrathin sections of segment C1. (**D**) 3D reconstruction of the dendritic segment C1. The FS basket cell axon (red) did not establish a synaptic contact with the dendritic segment C1 (red arrow). Scale bar in (**B**) is the same for (**D**).

The following figure supplement is available for figure 3:

**Figure supplement 1**. Focus step images for C1 dendritic segment with FS cell axonal fiber contact site shown in *Figure 2B*.

0.13 mV at the soma (*Figure 8D,F,K*). The spine head Sp1 was hyperpolarized without attenuation (*Harnett et al., 2012*), while the D1 synapse reached only 30% of the Sp1 synapse peak membrane potential. The peak synaptic current was 2.45 pA at the spine head, and 1.55 pA at the soma (*Figure 8E*). Injecting a synaptic waveform of 0.71 nS at the soma (S1) (*Table 2*) hyperpolarized that site by 0.48 mV (*Figure 8G, H*) resulting in an IPSC of 8.29 pA (*Figure 8H*), in the range of background noise. Simultaneous activation of somatic contacts S1–S4 resulted in a hyperpolarization of 1.33 mV, corresponding to a somatic current of 22.67 pA, (*Figure 8I,J*) similar to IPSP amplitudes from paired recordings of FS basket cells to hippocampal pyramidal cells (0.5–3 mV) (*Buhl et al., 1994*) and our own data (*Figure 2E*, *Table 5*). Thus for a similar driving force, proximal inhibitory synapses produce larger somatic hyperpolarizations than distal ones (*Figure 8K*).

Spines innervated by inhibitory synapses are typically excited by thalamic inputs (*Kubota et al., 2007*). We modeled the Sp1 spine to ask how spine-head IPSCs affect these excitatory thalamic signals (*Gulledge et al., 2012*). Excitatory synaptic events (0.2 nS) were greatly reduced by a coincident spine-head IPSC (*Figure 8L*). Excitation of the spine-head site depolarized the pyramidal cell soma by 0.12 mV. Simulated release from four somatic inhibitory synaptic sites hyperpolarized the soma by 1.33 mV. Thus inhibition from clustered somatic synapses of one FS basket cell effectively suppressed dendro-somatic conduction of inputs from ~11 excitatory spine synapses. If release probability depends on terminal size (*Holderith et al., 2012*), then GABA may be infrequently liberated from smaller inhibitory terminals made by FS basket cells at dendritic sites. Since inhibitory synapses from a single cell usually contact different, distant dendrites, resulting hyperpolarizations may sum poorly (*Figure 9*). Even so, summation of integrated dendritic signals during inhibitory cell firing at frequencies of 40–50 Hz (*Isomura et al., 2009*) together with GABAergic shunting effects (*Gidon and Segev, 2012*) may permit FS cell synapses to suppress excitatory inputs on innervated dendritic branches (*Cossart et al., 2001*). Diffusely located inhibitory terminals on dendritic shafts can therefore effectively control afferent excitatory signals.

Variation in release from single synaptic boutons contributes to event-by-event fluctuations in postsynaptic currents (*Sasaki et al., 2012*). IPSC amplitude varied substantially between trials in all dual recordings (*Figure 8N,O*, *Table 5*). Monte Carlo simulations were made on the model of pair CS56 to ask whether this variability might result from probabilistic IPSC generation at somatic terminals, S1–S4 (*Figure 8P*). Mean IPSC charge transfer was 193.1 fC ± 56.2 (89.9–326.1 fC, n = 60 traces; *Table 5*), with putative electric charge at somatic synapses calculated by multiplying junctional size by unit electrical charge, S1–S4 to give 120.1, 59.7, 66.6 and 79.6 fC respectively (*Table 2*). Release probability (0.59) was obtained by dividing the average electrical charge, 193.1 fC, by the maximum charge, 326.1 fC (*Table 5*). Somatic synapses were activated randomly with release probabilities correlated with junctional area (S1: 0.8. S2: 0.4, S3: 0.45, S4: 0.55) (*Figure 8—figure supplement 1*) (*Holderith et al., 2012*). IPSC charge distributions from paired recordings and simulations were statistically similar (p = 0.41 Kolmogorov–Smirnov, *Figure 8N,P*), suggesting that IPSC amplitude

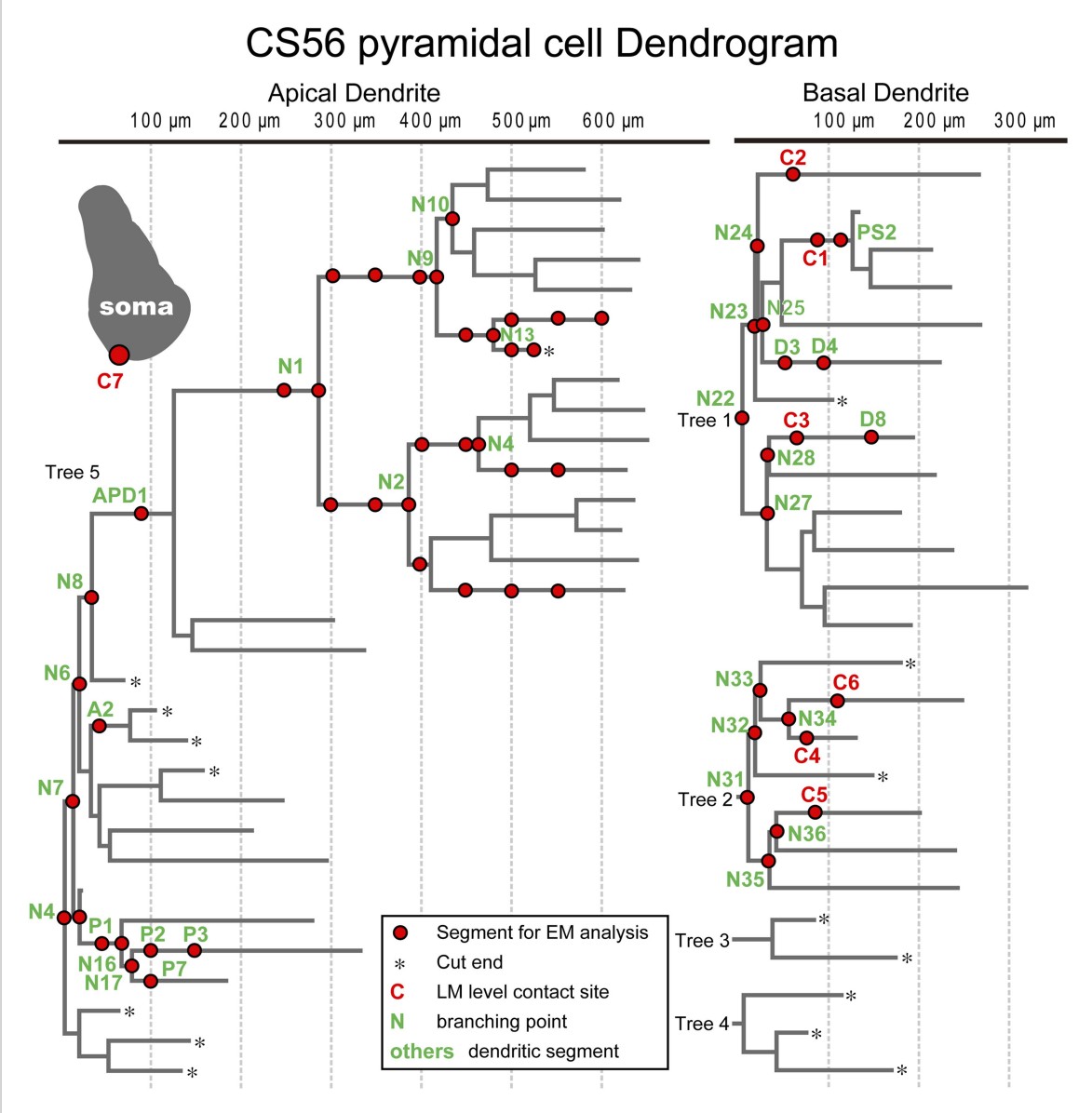

**Figure 4**. Dendritic segments and the somatic region selected for further quantitative EM analysis. Dendrogram of the apical (left) and basal (right) dendrites of the postsynaptic pyramidal cell of pair CS56. Dendritic segments indicated by red circles and numbers and the somatic region (inset grey drawing) were selected and analyzed in serial ultrathin sections at the EM level. In this pair seven synaptic contact sites were identified at the light microscopic level (C1–C7).

variations result from an independent, stochastic activation of individual somatic and proximal synapses (*Sasaki et al., 2012*).

## General principle of cortical inhibitory connections

We suggest that FS cell inhibitory synaptic strength is progressively reduced from terminals contacting the soma to dendritic shafts and then spines of target pyramidal cells. We asked whether this represents a general principle for cortical inhibitory connections by comparing synapses made by different classes of cortical interneurons stained using the whole cell recording method (*Figure 10A*) (*Kubota et al., 2007*). 3D reconstruction of serial EMgs let us calculate synaptic junction area and the cross sectional area of postsynaptic dendrite or spine volume, for 305 synapses made by 9 different types of interneuron. The junctional area of somatic inhibitory synapses was 0.40 ± 0.15 μm² (n = 23),

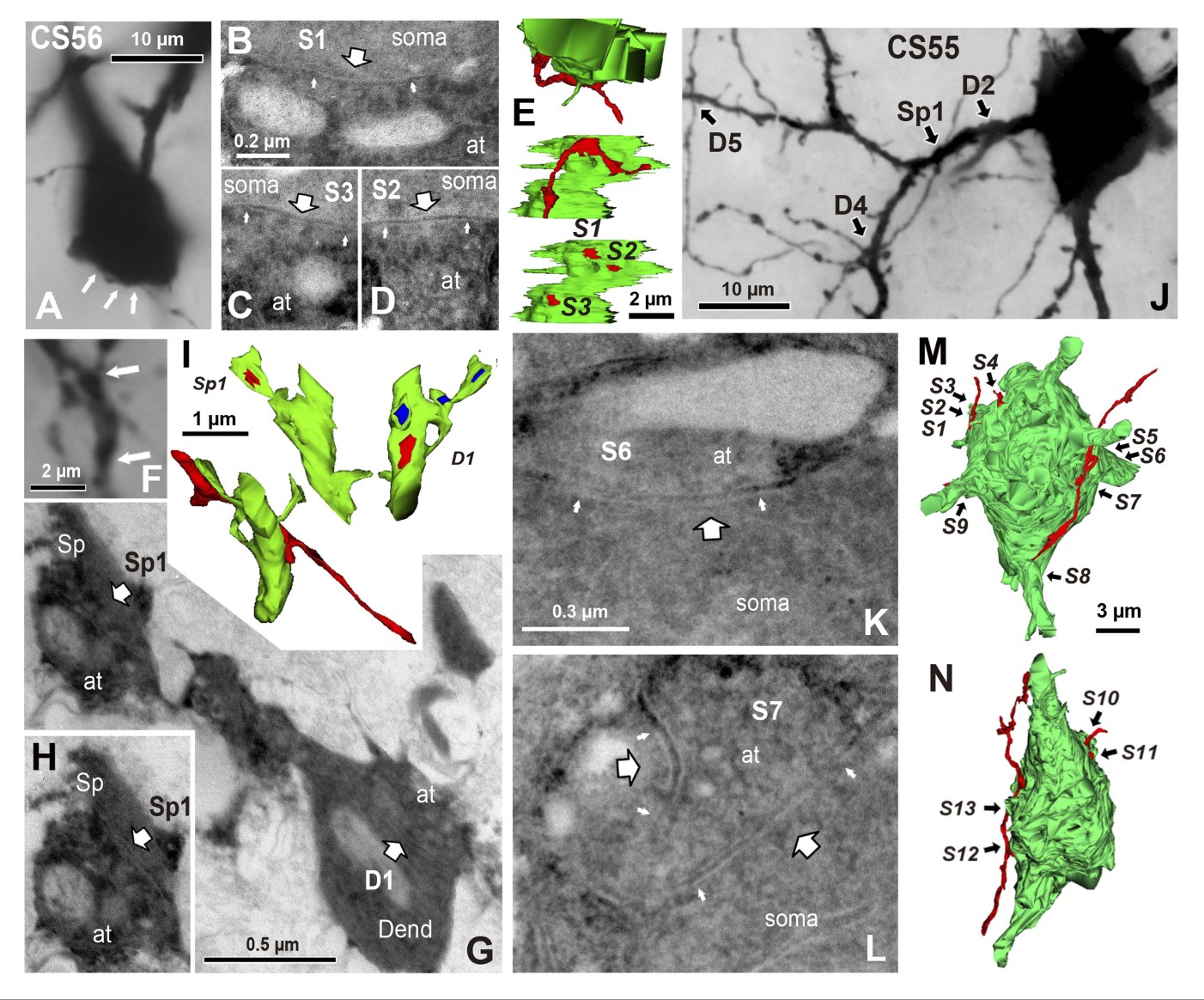

**Figure 5**. EM identification of synaptic contacts. (**A**) LMg of putative synaptic contacts (white arrows) established by a basket cell axon on the soma of a pyramidal cell of CS56. (**B–D**) EMgs of three somatic synaptic contacts (S1–S3). Thick arrows indicate synaptic junctions, small arrows the extremities of the synaptic cleft. (**E**) The upper view is a 3D reconstruction of somatic synapses (red) on the soma (green) in the same plane as in (**A**), the middle image, rotated by 90°, shows three boutons apposed to the pyramidal cell soma and the lower view shows their synaptic junctions. (**F**) LMg of putative synaptic contacts on a pyramidal cell dendrite. (**G**) EMg of synapses with a dendritic spine (Sp1, upper left arrow) and dendritic shaft (D1, bottom right arrow) 40° tilting angle. (**H**) EMg of the spine synapse in **G** (arrow). (**I**) 3D reconstructions of the synapses in (**G**). Lower left image shows the dendritic segment indicated by arrows in (**F**). Middle view, rotated by ~60°, shows the junction made with the spine (red). Right image is rotated by ~ −90° to visualize the junction on the dendrite. (**J**) Focus stack image of LMg of putative contacts (arrows) made by basket cell axonal terminals on a pyramidal cell soma and dendrites of CS55. (**K**, **L**) EMgs of the S6 (**K**) and S7 (**L**) somatic junctions. (**M**, **N**) Two views of a 3D reconstruction of a FS cell axon (red) and pyramidal cell soma (green) showing all contacts. (at, axon terminal; sp, spine; dend, dendrite).

The following figure supplements are available for figure 5:

**Figure supplement 1**. Somatic synapse contact sites identified using electron microscopic observation.

**Figure supplement 2**. Focus step images for CS55 pair neurons shown in *Figure 3J*.

**Figure supplement 3**. The presynaptic FS basket cell axon terminal crosses the postsynaptic pyramidal cell CS55 dendrite.

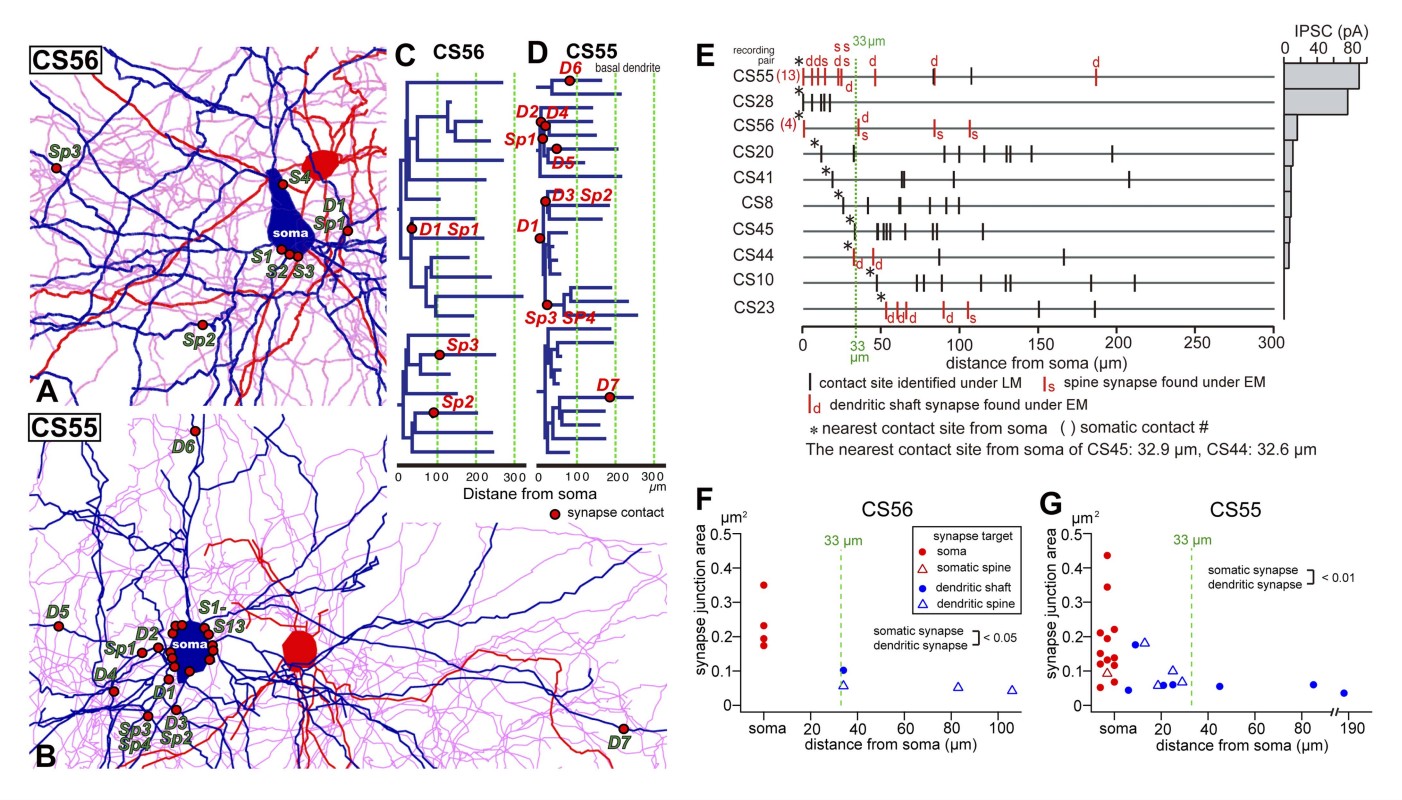

**Figure 6**. Synapse contact sites identified by EM observation of pairs CS56 and CS55. (**A**, **B**) Synaptic contact sites are shown in drawings of CS56 pair neurons (**A**) and CS55 pair neurons (**B**). Postsynaptic pyramidal cell soma and dendrites are in blue, presynaptic FS basket cell soma and dendrites are in red, and axon in pink. (**C**, **D**) The synapse contact sites are shown in dendrograms of the basal dendrites of postsynaptic pyramidal cell of CS56 pair (**C**) and CS55 pair (**D**). (**E**) Distribution of putative synaptic contacts (black bars) made by single basket cells on somato-dendritic membrane of 10 pyramidal cells. Contacts confirmed by EM are shown in red. (**F**, **G**) Area of somatic synaptic junctions is significantly larger than those on dendritic shafts and spines of CS56 pair neurons (**F**) and CS55 pair neurons (**G**).

The following figure supplement is available for figure 6:

**Figure supplement 1**. Linear correlation of synapse junction area and bouton volume.

for dendritic shaft synapses it was $0.19 \pm 0.12$ $\mu m^2$ (n = 195) and for synapses terminating on spines it was $0.09 \pm 0.05$ $\mu m^2$ (n = 87). Synaptic junctional area was therefore correlated with the size of the target structure (**Figure 10B–L**) with the possible exception of Martinotti cell terminals (**Figure 10J**) that contact distal pyramidal cell dendrites (**Silberberg and Markram, 2007**). Linear relations between synapse junction and post-synaptic target size (**Figure 10B–L**) may provide an effective impedance matching (**Kubota and Kawaguchi, 2000**) and thus control the inhibitory efficacy at different sites. Thus the variation in effects of FS basket cell synapses targeting different membrane regions on L5 pyramidal cells may reflects a general principle for inhibitory cortical circuits.

## Discussion

These data show that FS basket cells mediate either a global somatic inhibition of variable strength, a local dendritic shaft inhibition or act as a local veto at single spines. These distinct effects depend on differences in junctional size. Local spine or shaft potential changes are small and locally restricted. In contrast, somatic inhibitory currents are large, and summation of events from several somatic terminals produces a global control of pyramidal cell excitation. Somatic junctions have large areas, suggesting high release probability (**Holderith et al., 2012**) and typically contact multiple sites (**Buhl et al., 1994**). This enhances the likelihood of simultaneous release as FS cells fire repetitively at 30–50 Hz during motor behaviors in vivo (**Isomura et al., 2009**). Some FS basket cell connections with

**Table 2**. Synapse properties of pair CS56

| Synapse | Target | junction area (μm²) | Electric charge (fC)* | Conductance (nS)† | Distance from soma (μm) |
|---------|--------|---------------------|----------------------|-------------------|-------------------------|
| S1 | Soma | 0.350 | 120.1 | 0.71 | 0 |
| S2 | Soma | 0.174 | 59.7 | 0.35 | 0 |
| S3 | Soma | 0.194 | 66.6 | 0.39 | 0 |
| S4 | Soma | 0.232 | 79.6 | 0.47 | 0 |
| Sub total | | 0.950 | | | |
| D1 | Dendrite | 0.102 | 35.2 | 0.21 | 34 |
| Sp1 | Spine | 0.056 | 19.2 | 0.11 | 34 |
| Sp2 | Spine | 0.051 | 17.6 | 0.10 | 83 |
| Sp3 | Spine | 0.042 | 14.4 | 0.08 | 106 |

*Estimated from junctional area.
†Estimated from electric charge.

pyramidal cells involved exclusively dendritic sites while others consisted of both peri-somatic and proximal dendritic contacts. Spines receiving inhibitory synapses are typically large (*Kubota et al., 2007*) and their thalamic excitatory inputs presumably express both NMDA and AMPA receptors (*Matsuzaki et al., 2004*; *Kubota et al., 2007*). Inhibitory synapses may then efficiently veto these thalamic inputs before activation of NMDA receptors (*Gulledge et al., 2012*) so reducing the probability of pyramidal cell firing.

In paired recordings IPSCs were detected only for terminals that contacted proximal pyramidal cell dendrites. However, IPSCs initiated on distal dendrites have been recorded at the soma in some studies (*Silberberg and Markram, 2007*; *Jiang et al., 2013*). Possibly differences in experimental paradigm are responsible. In this work post-synaptic potentials were more hyperpolarized (−65 mV rather than −55/−57 mV) and Cl⁻ in the recording pipette was higher (19 rather than 10 mM) than in other studies. Both differences would encourage somatic propagation of IPSPs initiated at distant dendritic sites. In our somatic recordings we did not detect IPSPs generated at synapses more distant than ∼40 μm. Possibly, the Cl⁻ reversal potential was similar to the holding potential resulting in a small or null driving force at these sites. Indeed unperturbed Cl⁻ reversal potentials may be 10–25 mV more hyperpolarized than in invasive whole-cell recordings (*Verheugen et al., 1999*; *Bevan et al., 2000*). Further work is needed to define unperturbed Cl⁻ reversal potentials in the dendrites and soma of L5 pyramidal cells.

Distinct numbers and sites of synaptic contacts made by FS interneurons with pyramidal cells may be regulated by network function (*Yoshimura et al., 2005*) and activity during different states (*Klausberger and Somogyi, 2008*; *Puig et al., 2008*). The strength of inhibition mediated by hippocampal FS basket cells varies with different target pyramidal cells. Synaptic strength is greater at connections with CA1 pyramidal cells in deep rather than superficial layers of stratum pyramidale (*Lee et al., 2014*) and it is genetically coded (*Donato et al., 2015*). The innervation patterns of cortical basket cells appear to be regulated by experience, environment or fear conditioning (*Donato et al., 2013*), according to network properties (*Yoshimura et al., 2005*; *Lee et al., 2014*) and the activity in specific target cells (*Xue et al., 2014*), and activity level of them may be regulated by learning as well as genetics (*Donato et al., 2015*). In contrast, the efficacy of synapses made by Martinotti cells seems to be independent of target pyramidal cell activity (*Xue et al., 2014*). Thus different cortical interneurons respond in distinct ways to neuronal network state.

The size, and thus efficacy, of synaptic terminals made by FS interneurons with the soma, dendritic shafts and spines of target pyramidal cells were measured from 3D EM reconstructions. Other GABAergic interneurons establish domain-specific contacts (*Kawaguchi and Kubota, 1998*; *Jiang et al., 2013*; *Kubota, 2014*; *Marlin and Carter, 2014*). Paired recordings from other cortical interneurons and pyramidal cells followed by complete reconstruction of terminals will be needed to establish rules relating terminal size to efficacy. Nevertheless a somato-dendritic gradient of inhibitory terminal size may be a general principle. Our data suggests that relations between post-synaptic site, terminal properties including junctional area, and GABA release patterns may be maintained for other types of cortical interneurons.

**Table 3**. Synapse properties of pair CS55

| Synapse | Target | Junction area (μm²) | Electric charge (fC)* | Distance from soma (μm) |
|---------|--------|---------------------|------------------------|--------------------------|
| S1 | Soma | 0.116 | 40.9 | 0 |
| S2 | Soma | 0.221 | 77.6 | 0 |
| S3 | Soma | 0.052 | 18.4 | 0 |
| S4 | Soma | 0.120 | 42.3 | 0 |
| S5 | Soma | 0.436 | 153.0 | 0 |
| S6 | Soma | 0.194 | 68.2 | 0 |
| S7 | Soma | 0.344 | 121.0 | 0 |
| S8 | Soma | 0.151 | 52.9 | 0 |
| S9 | Soma | 0.068 | 23.8 | 0 |
| S10 | Soma | 0.138 | 48.3 | 0 |
| S11 | Soma | 0.132 | 46.2 | 0 |
| S12 | Soma | 0.211 | 74.1 | 0 |
| S13 | Somatic spine | 0.092 | 32.3 | 0 |
| Sub total | | 2.274 | | |
| D1 | Dendrite | 0.044 | 15.3 | 6 |
| D2 | Dendrite | 0.176 | 61.8 | 8.6 |
| Sp1 | Spine | 0.180 | 63.2 | 12.6 |
| D3 | Dendrite | 0.058 | 20.3 | 22.6 |
| Sp2 | Spine | 0.054 | 19.1 | 22.6 |
| D4 | Dendrite | 0.060 | 21.1 | 24.7 |
| Sp3 | Spine | 0.099 | 34.6 | 24.9 |
| Sp4 | Spine | 0.067 | 23.4 | 24.9 |
| Sub total | | 3.011 | | |
| D5 | Dendrite | 0.055 | 19.3 | 44.8 |
| D6 | Dendrite | 0.060 | 21.1 | 84.5 |
| D7 | Dendrite | 0.046 | 16.2 | 188.5 |

*Estimated from junctional area.

Inhibitory synapses terminating on spines form 25–50% of GABAergic contacts with cortical pyramidal cell (*Kubota et al., 2007*; *Chen et al., 2012*) and so form a major part of inhibitory microcircuits. Spines contacted by an inhibitory synapse are typically co-innervated by an excitatory thalamic input (*Kubota et al., 2007*). Our simulations show single inhibitory synapses can effectively veto synaptic excitation and intercept NMDA current (*Gulledge et al., 2012*; *Harnett et al., 2012*; *Chiu et al., 2013*) at the spine head. They could then prevent summation of thalamic excitatory inputs arriving within about 20 ms (*Marlin and Carter, 2014*), as pyramidal cell and FS basket cells are co-activated by thalamo-cortical afferents (*Kimura et al., 2010*). Hence the FS basket cell acts as a feed forward inhibition to thalamic input.

Excitatory synapses innervating cortical pyramidal cell spines can be modulated by visual experience (*Chen et al., 2012*) or by somatosensory stimulation (*Knott et al., 2002*). The veto by inhibitory synapses terminating on spines may be especially important for such plastic changes (*Chen et al., 2012*). Pyramidal cell dendritic spines are tuned to distinct modalities and spines with similar preferences may not cluster together on the same dendritic branch but averaged across a neuron biased towards the orientation tuning of the cell's output (*Chen et al., 2013*). Inhibitory synapses on dendritic shafts may then inhibit tuned/untuned excitatory inputs on the same but not different dendritic branches and so efficiently and specifically adjust pyramidal cell activity (*Liu, 2004*; *Marlin and Carter, 2014*). Our data shows dendritic IPSCs may exert strictly local effects. Cl⁻ reversal potential at distal dendrite/spine synapses may normally be close to the local resting membrane potential. However this small driving

**Table 4**. Proportion of basket terminal

| Pair | Basket terminal | Total bouton | Basket terminal (%) |
|---|---|---|---|
| CS55 | 106 | 285 | 37.2 |
| CS28 | – | – | – |
| CS56 | 91 | 211 | 43.1 |
| CS20 | 52 | 217 | 24.0 |
| CS41 | 73 | 248 | 29.4 |
| CS8 | 59 | 201 | 29.4 |
| CS45 | 26 | 151 | 17.2 |
| CS44 | 67 | 233 | 28.8 |
| CS10 | 59 | 226 | 26.1 |
| CS23 | 63 | 315 | 20.0 |
| Total/average | 596 | 2087 | $28.4 \pm 7.6$ |

force would be increased by depolarization due to dendritic EPSPs. IPSPs will then reduce EPSP amplitude at the soma even if they do not propagate somatically. FS cells can thus control excitation of L5 pyramidal cells by a specific, local veto of co-innervated spines, by reducing dendritic propagation of summed EPSPs as well as by a strong, global peri-somatic inhibition.

We have estimated a peak amplitude of $5.7 \pm 3.1$ pA for EPSCs generated at single synaptic contacts with CCS pyramidal cell proximal dendrites (*Morishima et al., 2011*). Here we found a peak IPSC amplitude of 2.4 pA at dendritic shaft synapses. Our simulations suggest that summation of single excitatory and inhibitory synaptic currents may reduce dendritic excitation and suppress calcium entry via NMDA receptors (*Larkum et al., 2009*). GABA$_A$ receptor activation will also reduce EPSP amplitude by shunting (*Hao et al., 2009*; *Gidon and Segev, 2012*). Thus, activation of a single

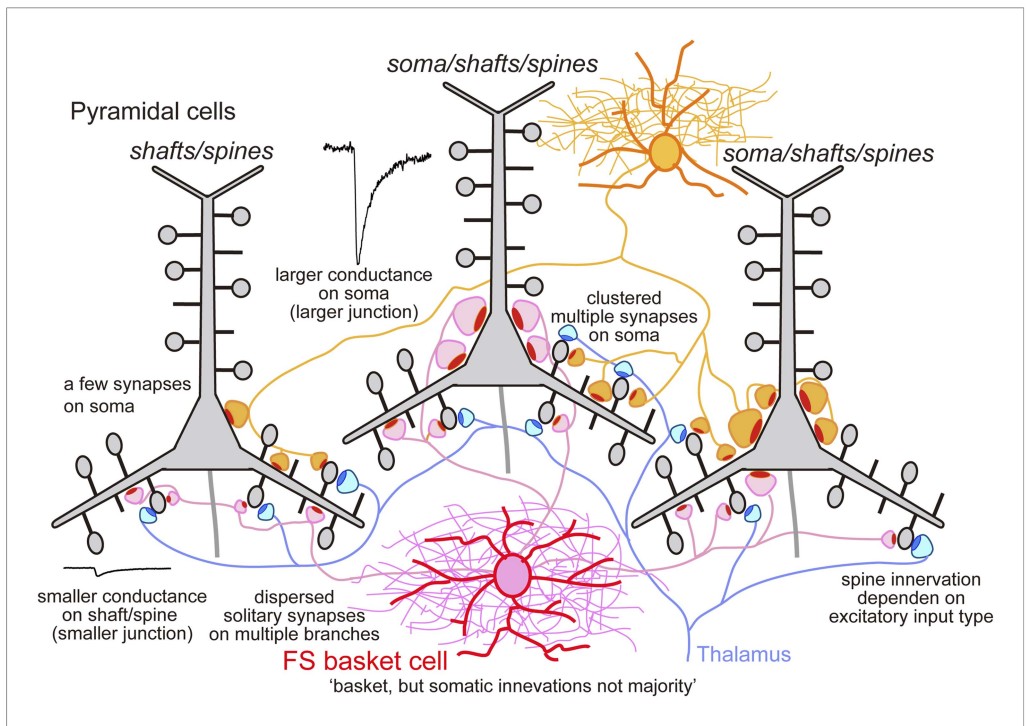

**Figure 7**. Schematic summary. Schematic drawing to summarize our main findings.

**Table 5**. IPSC properties of pair CS56 and CS55

| | CS56 | | CS55 | |
| --- | --- | --- | --- | --- |
| | Electric | peak | Electric | peak |
| | Charge (fC) | (pA) | Charge (fC) | (pA) |
| Average | 193.1 | −17.3 | 895.2 | −91.3 |
| SD | 56.2 | 3.0 | 96.2 | 11.2 |
| Max | 326.1 | −24.9 | 1057.8 | −111.0 |
| Min | 89.9 | −11.8 | 766.0 | −74.0 |
| n | 60 | 60 | 10 | 10 |
| Average Trace | 217.5 | −14.2 | 994.6 | −89.4 |

dendritic inhibitory synapse should effectively suppress EPSCs at nearby excitatory synapses. This distal dendritic inhibition is functionally strong (*Cossart et al., 2001*; *Gidon and Segev, 2012*). Inhibitory synapses on dendrites and spines act to reduce neuronal excitability by blocking local EPSCs and so decrease the amplitude of summed EPSPs. The synchronization of FS basket cell activity via gap junctions (*Gibson et al., 1999*) will further counter the summation of afferent EPSPs.

It is generally accepted that synaptic contacts detected by LM must be confirmed with EM. We verified 14 synapses of 25 putative dendritic contacts with LM (56%) in this study and 78% in our previous study (*Karube et al., 2004*). In addition, we newly found 6 dendritic/spine synapses with EM (30%; 6/20). Care must also be taken with somatic inhibitory terminals which are much smaller than the soma, so that terminals behind or in front of a soma may be impossible to resolve in LM. Indeed, we identified 14 somatic synapses with EM for CS55 and 4 somatic synapses with EM for CS56, although our estimation of the contacts with LM was three for the CS55 and one for the CS56 pair. Our data shows the importance of EM data for quantitative measurements on the number and size of synaptic junctions.

Passive cable properties and voltage-dependent resting conductances affect IPSP amplitude. Since postsynaptic target size is related to input resistance and synaptic junction area to the number of post-synaptic receptors (*Nusser et al., 1997*), alterations in synaptic dimensions may govern the size of GABAergic currents. The dependence of synaptic terminal areas on postsynaptic dendritic cross sectional areas would tend to maintain a constant ratio of synaptic conductance to post-synaptic input resistance. Thus, presynaptic interneuron actions are efficiently regulated to provide an appropriate hyperpolarization of their post-synaptic target (*Kubota and Kawaguchi, 2000*).

EPSC amplitude is correlated with synaptic junction area, release probability, calcium entry and receptor number (*Holderith et al., 2012*). At inhibitory synapses, currents are also correlated with release probability, docking site number and receptor number (*Nusser et al., 1997*; *Pulido et al., 2015*). Synaptic junctional area should then govern IPSC amplitudes. Surprisingly unit IPSCs from recordings in this work were quite similar, suggesting that the inhibitory synaptic current is well correlated with synaptic junction area. Larger synapses may generate larger IPSCs, due to multiple release sites or higher numbers of post-synaptic receptors. The presence of multiple release sites at some synaptic junctions has been shown by anatomy (*Holderith et al., 2012*; *Nakamura et al., 2015*) or estimated from neurophysiological data (*Nakamura et al., 2015*; *Pulido et al., 2015*). Clusters of the Cav2.1 Ca-channels in large synaptic junctions have been correlated with estimates of the number of vesicular docking sites. GABA release from multiple sites in a large synapse could saturate post-synaptic receptors and initiate large synaptic currents of similar amplitude, as at single-terminal synaptic connections made by molecular layer interneurons of the cerebellum. In contrast, the IPSCs

**Table 6**. Unit IPSC

| Pair | Electric charge (fC) | Junction area (μm$^2$) | Unit IPSC (fC/μm$^2$) |
| --- | --- | --- | --- |
| CS56 | 326.1 | 0.950 | 343.3 |
| CS55 | 1057.8 | 3.011 | 351.3 |

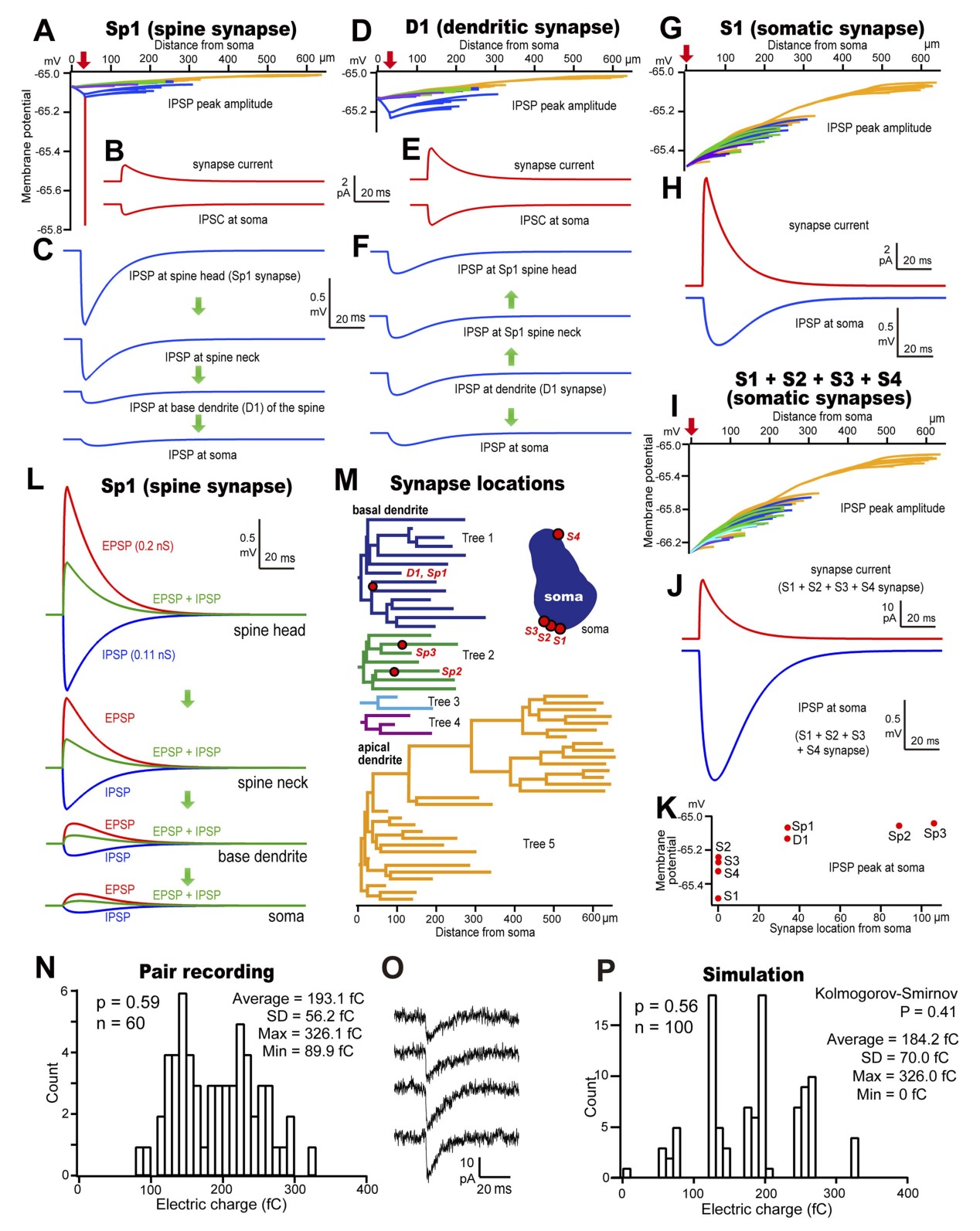

**Figure 8**. Simulated conduction for dendritic spine, shaft and somatic IPSCs. (**A–C**) Dendro-somatic conduction of a spine synapse IPSC. (**A**) Peak membrane potential changes (color-coded as in (**M**)) over somato-dendritic membrane induced by an IPSC of 0.11 nS injected at Sp1 of the model pyramidal cell (red arrow). Peak inhibitory potential of the spine in red. (**B**) IPSC waveform injected at Sp1 spine head is reduced to 64% at the soma.

*Figure 8. continued on next page*

*Figure 8. Continued*

(**C**) Simulated IPSPs. Current flow indicated by arrows. IPSP attenuation was 15% at the basal dendrite and 9% at the soma. (**D–F**) Conduction of a dendritic shaft IPSC, D1. (**D**) Peak somato-dendritic potential changes induced by an IPSC of amplitude 0.21 nS injected at a dendritic shaft (red arrow). (**E**) IPSC waveform injected at D1 (upper) and simulated somatic IPSC (lower trace) with an attenuation of 63%. (**F**) IPSP wave form. Current flow indicated by arrows. IPSP attenuation at the soma is 57%, but no attenuation at the spine. (**G**, **H**) Conduction of a somatic IPSC, S1. (**G**) Peak somato-dendritic potential changes induced by an IPSC of amplitude 0.7 nS injected at the S1 somatic site (red arrow). (**H**) IPSC waveform injected at S1 (upper) resulting in a somatic IPSP (lower). (**I**) Somato-dendritic conduction of the IPSC resulting from activating (red arrow) four somatic synapses S1, S2, S3 and S4. (**J**) Summed IPSC waveform (upper trace, S1–S4) and somatic IPSP (lower). (**K**) Peak somatic IPSPs for eight different injected IPSCs. (**L**) Reduction (green) of the EPSP resulting from the injection of an EPSC waveform of 0.2 nS (red) at the spine head, Sp1, by an IPSC (blue) injected at the same site and time. (**M**) Color-coded dendrogram and corresponding somatic synaptic contacts on the model cell. (**N**) Bar histogram showing the distribution of IPSC electric charge of the pair CS56. (**O**) IPSC variance of the pair CS56. (**P**) Bar histogram of the distribution of IPSC electric charge when simulated. Here, the IPSC electric charge also substantially varied from trial to trial and is not significantly different as in the paired recording (Kolmogorov–Smirnov test, p = 0.41).

The following figure supplement is available for figure 8:

**Figure supplement 1**. Relationship showing synapse conductance and release probability used for simulation analysis in *Figure 6P*.

examined here were mediated by multiple synaptic contacts of FS basket cells on L5 CCS pyramidal cells. IPSC amplitude fluctuations presumably reflected variations and failures in release from different terminals.

Axons of cortical non-pyramidal cells project to distinct laminar and columnar zones (*Kubota, 2014*), enabling different subtypes of interneurons to form synapses with specific targets. Projecting to a specified zone, an axon could make contacts nonspecifically with any available target neuron (*Fino and Yuste, 2011*; *Packer and Yuste, 2011*; *Packer et al., 2013*). Alternatively synaptic contacts may be established preferentially with specific neuronal subtypes or target domains, such as soma, axon or dendrites (*Jiang et al., 2013*). Target preference may depend on an activity dependent control of excitatory and inhibitory synaptic input size in order to maintain E/I balance (*Xue et al., 2014*). Our data show FS basket cells may form synaptic contacts with the perisomatic region of post-synaptic pyramidal cells or with their proximal dendritic shafts and spines. Inhibitory synaptic junctional area was matched to the synaptic site—it was larger at somatic than dendritic sites and larger at synapses made with shafts than at those made with dendritic spines. Molecular cues to recognize a somatic or dendritic innervation site may include chemoattractive and cell adhesion molecules. Such mechanisms are involved in a segregation of dendritic spine inhibitory inputs and distinct sources of afferent excitation. Spines innervated by FS basket cell terminals also receive excitatory synapses from thalamus, but never recurrent cortical pyramidal cell inputs (*Kubota et al., 2007*). Both activity dependent chemoattractant factors (*Yee et al., 1999*) and cell adhesion molecules of the protocadherin family (*Meguro et al., 2015*; *Yagi, 2015*) have been linked to this specificity. Functionally it would permit FS cell inhibitory synapses to mediate an efficient and selective veto on excitatory inputs from the thalamus.

A recent modeling paper (*Gidon and Segev, 2012*) enhanced our understanding of dendritic inhibitory operations. It assumed that inhibitory synapses targeting pyramidal cell somata, dendritic shafts and dendritic spines possess a uniform size, and strength. Our data suggests the model could be refined to explore the effects of variation in synaptic size and strength from soma to dendrite spine. Quantitative 3-D EM reconstructions provide an exact basis to assign different weights to inhibitory synapses that contact different sites. This inhibitory synaptic machinery differs from that at excitatory synapses subject to both plasticity (*Matsuzaki et al., 2004*) and scaling functions (*Magee, 2000*; *Katz et al., 2009*). Defects in these microcircuits may contribute to depression and other neuronal diseases (*Sauer et al., 2015*). Our data thus provide novel insights into biophysical design principles for inhibitory synaptic operations in neural microcircuits.

## Materials and methods

### Retrograde labeling of CCS cells

Retrograde labeling of CCS cells was performed as described previously (*Morishima and Kawaguchi, 2006*). Briefly, young Wistar rats (between postnatal 19–23 days old; Charles River, Japan)

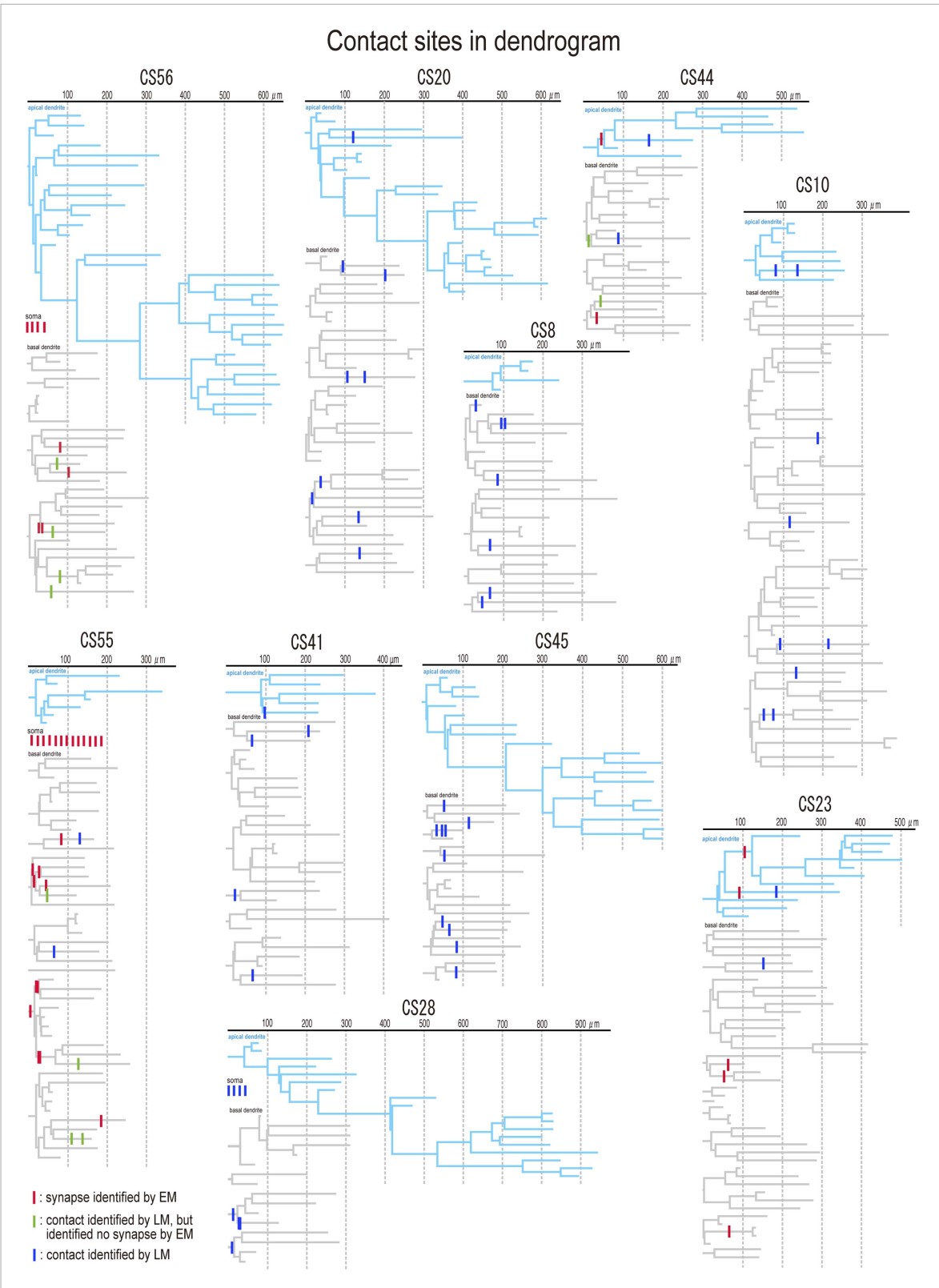

**Figure 9**. Dendrograms with contact sites of the post synaptic pyramidal cells. Individual dendrograms of all investigated postsynaptic pyramidal cells (n = 10). Apical dendrograms are shown in blue and basal dendrograms are in gray.

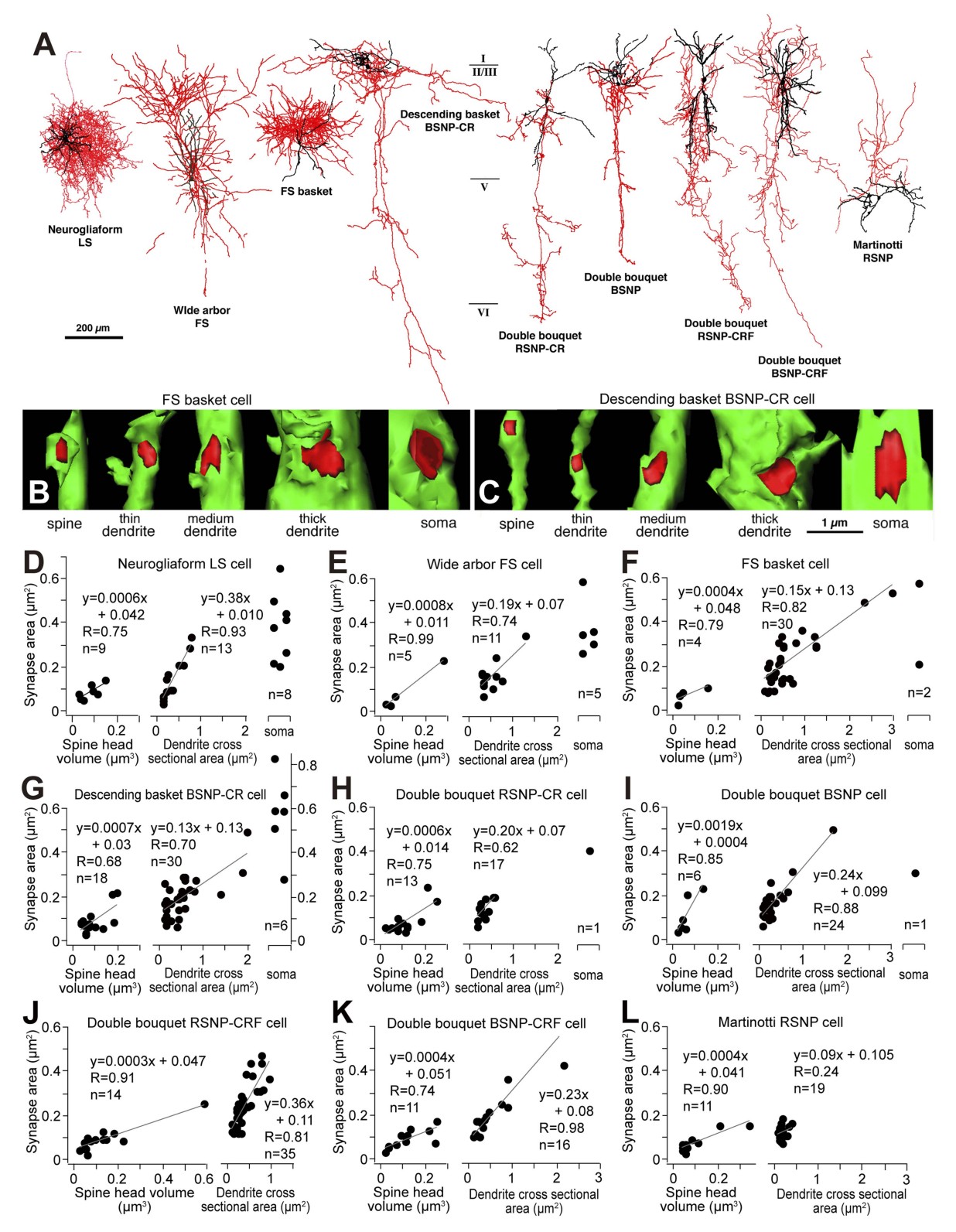

**Figure 10.** Linear correlation between synapse junction area and postsynaptic target size of non-pyramidal cells. (**A**) Different types of cortical GABAergic non-pyramidal cells. The somatodendritic domain of the neurons is given in black and their axons in red. Abbreviations: LS, late spiking cell; FS, fast spiking cell; BSNP, burst spiking non-pyramidal cell; RSNP, regular spiking non-pyramidal cell; CR, calretinin; CRF, corticotropin releasing factor. (**B**, **C**) 3D

*Figure 10. continued on next page*

*Figure 10. Continued*

reconstructions of synaptic junctions (red) on target structures (green) of inhibitory axon terminals by cortical FS basket cell (**B**) and descending basket BSNP-CR cell (**C**) using 3D serial EMgs. The thickness of the target structure (from left to right) is positively correlated with the size of the junction area. (**D–L**) Line diagrams correlating synaptic junction area of the non-pyramidal neurons with spine head volume (left panel), dendrite cross sectional area (middle panel) and plots with soma (right panel). The synapse junction area on spines and dendrites is linearly correlated with the target size. The somatic synapse is larger when compared with dendritic and spine synapse.

were anesthetized with ketamine (40 mg/kg body weight) and xylazine (4 mg/kg body weight). Rats were placed in a stereotaxic frame and the skull on the injection hemisphere was partially removed and the cortex, hippocampus and fimbria caudal to the striatum were suctioned to prevent the spilling of dye into the cortex during injection. Cholera toxin subunit B conjugated with Alexa Fluor 555 (CTB-555; C34776, Invitrogen, NY) was used as the retrograde tracer (0.2% dissolved in distilled water). Injection site was determined by using stereotaxic coordinates (0.8 mm posterior to bregma, 2.5 mm lateral to the midline, depth 4 mm) and a glass pipette (tip diameter is around 100 micron) filled with CTB-555 was inserted to the striatum obliquely. Injection (80–100 nl) was performed using positive pressure from a pneumatic pico-pump (PV-820, World Precision Instrument, Sarasota, FL). After injection, the aspirated brain space was filled with a gel sponge (Spongel, Astellas Pharma Inc., Tokyo, Japan) immersed with saline and the skin was sutured. Rats recovered from surgery in the animal facility and were used for electrophysiological experiments at 2–3 days after the injection.

## Slice preparation

Rats were deeply anesthetized with isoflurane and were decapitated after the loss of all responses to tactile stimuli, such as pinching legs. Slices of frontal cortex (300 μm thick) were cut in ice-cold artificial cerebrospinal fluid ACSF with a vibratome (VT1000S, Leica, Germany) and kept at room temperature in ACSF until recordings. The ACSF consisted of (in mM) 124 NaCl, 3 KCl, 2.4 $CaCl_2$, 1.2 $MgCl_2$, 26 $NaHCO_3$, 1 $NaH_2PO_4$, 20 glucose, 0.4 ascorbic acid, 2 pyruvic acid and 4 lactic acid and saturated with 95%$O_2$/5%$CO_2$.

## Paired recordings

Slices were transferred to a recording chamber and perfused at 1–2 ml/min with ACSF (25°C). Patch pipettes (3–5 MΩ) were pulled from borosilicate glass and filled with 20 μl of internal solution containing (in mM): 126 K-methylsulfate, 6 KCl, 2 $MgCl_2$, 0.2 EGTA, 4 ATP, 0.3 GTP, 10 phosphocreatine, 10 HEPES and 0.75% biocytin. The pH of the pipette solution was adjusted to 7.3 with KOH and the osmolality was set to 295 mOsm. Potassium-methylsulfate as internal solution provided a physiological space clamp (*Fleidervish and Libman, 2008*). Dual patch-clamp whole-cell recordings (EPC9/dual, HEKA, Germany) were made in the frontal cortex (medial agranular and anterior cingulate cortex) with the use of × 40 water-immersion objective (Axioskop FS, Carl Zeiss, Germany). Series resistance was typically 6–15 MΩ and was not compensated. If it exceeded 20 MΩ, data were discarded. Liquid junction potential was not corrected. The data were recorded at 10 kHz and filtered at 2 kHz. For paired whole-cell recordings, retrogradely labeled pyramidal neurons were selected under fluorescence and differential interference contrast microscope (DIC) (*Stuart et al., 1993*). FS basket cells were identified in acute slices by their appearance under DIC microscopy. FS cells were recorded using the above internal solution, while pyramidal cells were recorded using an internal solution with the KCl concentration raised to 15 mM and K-methylsulfate lowered to 117 mM to depolarize the reversal potential of $Cl^-$ (−52.5 mV). IPSCs were recorded as inward currents at −65 mV holding potential. APs were initiated in the presynaptic neuron by 1 ms depolarizing pulses of 300 pA. Presynaptic APs and postsynaptic currents were recorded simultaneously.

## Electrophysiological data analysis (*Figure 1—figure supplement 2*)

Recorded presynaptic potentials and postsynaptic IPSCs were analyzed off-line with IGOR software (WaveMetrics, Lake Oswego, OR). For the calculation of kinetic parameters of postsynaptic currents, traces with spontaneous synaptic currents on the rising or decay phase were omitted. The onset of the postsynaptic current was estimated by fitting the rising phase with a parabola and extrapolating back

to the baseline. Postsynaptic current amplitude was measured as the difference between the peak current, measured from a 1.5 ms window centered at the peak, and the average baseline current, measured in a 4 ms window preceding the presynaptic AP. The decay time constant was obtained by fitting the decay phase of postsynaptic current with a double exponential equation. Since synaptic responses systematically run-down during the time course of some experiments, the amplitudes of postsynaptic currents were plotted against time and only stable periods were selected for further analysis. On average 100 traces (range 50–150) were analyzed for each experiment. Postsynaptic currents smaller than 2 times the noise level were discarded as failures, and the amplitudes of the remaining postsynaptic currents were analyzed. Cumulative histograms of postsynaptic current and noise were constructed and compared with a paired t-test and confirmed the separation between two (*Figure 1—figure supplement 2B*). To average postsynaptic currents, the peaks of the postsynaptic currents were aligned. The electric charge of IPSC was analyzed using AxoGraph (Molecular Devices, Sunnyvale, CA). Values are reported as mean $\pm$ standard deviation.

## Visualization of recorded cells

After electrophysiological recordings, slices were immersion-fixed (1.25% glutaraldehyde, 4% para-formaldehyde, 0.2% picric acid in 0.1 M phosphate buffer) and irradiated for 10 s using a microwave, and kept at room temperature for 2 hr. Slices were then cryoprotected with sucrose containing 0.1 M phosphate buffer (15% followed by 30% of sucrose solution) and freeze-thawed in the liquid nitrogen. Slices were re-sliced at 50 µm thickness with the vibratome and reacted with avidin-biotin peroxidase complex solution (ABC kit, Vector Laboratory, Burlingame, CA). Biocytin-filled cells were visualized with 3,3'-diaminobenzidine tetrahydrochloride (0.02%), nickel ammonium sulfate (0.3%), and $H_2O_2$ (0.004%). Slices were further post-fixed in 1% $OsO_4$ with 7% glucose, dehydrated and embedded in plastic (Epon 812 resin kit, TAAB, Aldermaston, UK) between silicone (Sigma coat, Sigma–Aldrich, St. Louis, MO) coated glass slide and cover slip.

## Morphological analysis

Axons, dendrites, and somata of stained neurons were reconstructed using the *Neurolucida* software (MBF Bioscience, Williston, VT) attached to a NIKON ECLIPSE microscope equipped with a 60× objective lens (NA 1.4, NIKON, Tokyo, Japan). Inter point interval of drawing axons and dendrites was less than 2 micron. No correction was made for tissue shrinkage, which should be about 90% (*Karube et al., 2004*). Putative synaptic contacts were identified and their location was marked on the traces of axons and dendrites. The software *Neuroexplorer* was used for morphometrical and quantitative analyses of reconstructed cells, including total dendritic length and distances between somata and putative synaptic contacts.

## Focus stack image

The dendritic segment or soma images of every 0.5 µm focus step in the same image field were captured using the *Neurolucida* software (MBF Bioscience, Williston, VT) attached to a NIKON ECLIPSE microscope equipped with a 60× objective lens (NA 1.4, NIKON, Tokyo, Japan) and CCD camera (1392 × 1040 pixels). The focus stack image was obtained using 'auto-blend layers/stack images' function of Photoshop (Adobe, San Jose), which combine the best focused area of the multiple focus step images, to give a greater depth of field (http://en.wikipedia.org/wiki/Focus_stacking).

## EM analysis

After reconstruction with *Neurolucida*, stained neurons were serially sectioned at a thickness of 50 nm with an ultramicrotome (Reichert Ultracut S, Leica Microsystems, Germany). Ultrathin sections were mounted on Formvar-coated single-slot grids. EM images of labeled axon terminals and dendrites were captured with a CCD camera (XR-41, Advanced Microscopy Techniques) in Hitachi H-7000, and HT-7700 EMs (Hitachi, Tokyo, Japan) at magnification ×8,000 or ×15,000. Structures of interest were reconstructed and quantified from the serial EM images, with the 3D reconstruction software, Reconstruct (http://synapses.clm.utexas.edu/tools/index.stm) (*Fiala, 2005*). The synaptic junctions were segmented at a typical cleft structure that was found between presynaptic vesicle aggregations and postsynaptic membrane density.

## Simulation analysis

Simulations were made with NEURON (*Hines and Carnevale, 1997*). The morphology of the model neuron was reconstructed from the EM imaging data. Pyramidal cell dendrites typically possessed elliptical cross sections, but NEURON is limited to circular morphologies. We circumvented this problem by first modeling the pyramidal neuron with circular dendritic cross sections, preserving the cross sectional area from EM. Then, leak conductance and membrane capacitance densities in each section in the circular model were adjusted to be equivalent to those predicted from EM imaging data. Our pyramidal model incorporates passive leak channels only. The passive leak conductance and membrane capacity before adjustment were 0.0001 S/cm$^2$ and 1 μF/cm$^2$, respectively. The intracellular resistance for somata, basal and apical dendrites was 100 Ωcm, and for the spine head and spine neck 385 Ωcm, respectively. The equilibrium potential of the leak current was set to −65 mV. As above, the passive leak conductance and membrane capacitance density in each section in the NEURON model were modified in order to mimic the elliptical shape (for further details, refer to our previous paper [*Kubota, et al., 2011a*]). The relationship between cross sectional area (S), circumference (L) and summed length of distal dendrites (R) we used here is (S) = 0.00033258(R) + 0.048097 and (L) = 0.0012661(R) + 1.3206.

The membrane potential was set to −65 mV (*Morishima and Kawaguchi, 2006*), and the GABA$_A$ reversal potential to −77.5 mV (*Gulledge and Stuart, 2003*) to fit our measurements of driving force. The electrical charge of each synaptic contact was calculated by multiplying the synapse junction area by the unit electrical charge; in turn individual synaptic conductance was calculated from the electric charge (*Table 2*). The synaptic current was adjusted to the average current of pair CS56 (*Figure 2E*, lower panel) with a double exponential fit. It was injected at sites where the presynaptic FS basket cell axon established synaptic contacts with the pyramidal cell.

A kinetic model was used for inhibitory synapses (*Destexhe et al., 1994*). Parameters were estimated by fitting the model to the unitary max IPSC data (*Figure 2E* upper panel). The estimated duration time, rise time constant, decay time constant and conductance are 2.3 ms, 0.45 ms, 14.17 ms and 1.92 nS, respectively. Individual synaptic conductance was estimated as multiplying 1.92 nS (conductance of the unitary max IPSC) by the ratio of synaptic junctional area of each synapse to the total area of the 4 somatic synaptic junctional area (0.950 μm$^2$). The values of synaptic conductance corresponding to contact sites, S1, S2, S3, S4, D1, SP1, SP2, SP3 are given in *Table 2*.

The release probability for the simulation of IPSC variation was estimated with modified fitting line of *Figure 4H* in *Holderith et al., 2012*), y = 3.271 * 0.68 + 0.018. We multiplied slope of the fitted line by 0.68 to get the similar release probability with pair cell recording result (*Figure 8—figure supplement 1*).

## Single cell electrophysiology experiment

Experiments were performed as described for the electrophysiological recording experiments previously (*Kubota et al., 2007*). Briefly, whole-cell access was obtained in neurons using visual DIC optics and a 40x water immersion objective. The pipette solution consisted of (in mM): potassium methylsulfate, 120; KCl, 5.0; EGTA, 0.5; MgCl$_2$, 1.7; Na$_2$ATP, 4.0; NaGTP, 0.3; HEPES, 8.5; and biocytin, 17. The recording was usually performed for 10–20 min. After re-slicing at 50 μm thickness, each slice (a set of 50 μm sections after resectioning) was further treated by one of the following two procedures.

(A) Some slices were incubated with avidin-biotin peroxidase complex (ABC) solution (Vector Laboratory, Burlingame, CA) in Tris–HCl buffered saline (TBS) with or without 0.04% Triton X-100 (TX), and reacted with 3,3-diaminobenzidine tetrahydrochloride (DAB) (0.05%) and H$_2$O$_2$ (0.003%) in 0.1 M phosphate buffer (PB).

(B) Other slices were processed for fluorescence immunohistochemistry to identify neurochemical markers, CRF and calretinin. The slices were incubated with the primary antibodies, CRF developed in rabbit (1:1000, gift by Dr. Wylie Vale, #PBLrC70) and calretinin (1:1000, Swant, Bellinzona, Switzerland, #6B3) in TBS containing 2% bovine serum albumin, 10% normal goat or horse serum and 0.04% TX. The slices were incubated in fluorescent secondary antibodies, followed by incubation with Alexa 350 streptavidin (1:200, Molecular Probes, Eugene, OR, #S-11249) in TBS. After examination for fluorescence, the slices were incubated with ABC, and reacted with DAB and H$_2$O$_2$.

Slices were then post-fixed in 1% $OsO_4$ in 0.1 M PB, dehydrated and flat embedded on silicon-coated glass slides in plastic (Epon 812 resin kit, TAAB, Aldermaston, UK). Recovered neurons were drawn using a drawing tube, or 3D reconstructed using the *Neurolucida* software (MBF Bioscience, Williston, VT) attached to a NIKON ECLIPSE microscope equipped with a 60× objective lens (NIKON, Tokyo, Japan). After light microscopic reconstruction, stained cells were serially sectioned into 90 nm thickness using an ultramicrotome (Reichert Ultracut S). Ultrathin sections mounted on one-hole grids were stained with lead citrate. Electron micrographs were taken with a Hitachi H-7000 electron microscope (EM), using tilting of up to 60˚. EM images of the labeled terminals and associated structures were captured using a CCD camera and reconstructed three-dimensionally (Visilog; Noesis, France).

## Statistics

We used Mann Whitney U test (non-parametric) to compare the junctional area of somatic and dendritic/spine synapses (*Figure 6F,G*) and Kolmogorov–Smirnov test to compare electric charge distributions from paired recordings experiment and the simulation of *Figure 8N,P*.

## Datasets

The datasets I can provide are Neurolucida reconstructed neuron to the "NeuroMorpho.Org", http://neuromorpho.org/neuroMorpho/index.jsp (*Kubota, 2015a*, *2015b*, *2015c*, *2015d*), and authentic model cell for 'Neuron' simulator to the 'ModelDB', https://senselab.med.yale.edu/modeldb (*Kubota, 2015e*).

## Acknowledgements

We thank H Kita and C Shiozu for technical assistance and Dr M Ushimaru for helping to analyze the physiological data. We thank Drs R Miles and JL Chen for comments on the manuscript. This study was supported by the EM facility in National Institute for Physiological Sciences in Japan.

## Additional information

### Funding

| Funder | Grant reference | Author |
|---|---|---|
| Ministry of Education, Culture, Sports, Science, and Technology (MEXT) | Grant-in-Aid for Scientific Research (B), 25290012 | Yoshiyuki Kubota |
| National Institutes of Natural Sciences | The Imaging Science Project of the Center for Novel Science Initiatives (IS261004) | Yoshiyuki Kubota |
| Toyoaki Scholarship Foundation | | Yoshiyuki Kubota |
| Uehara Memorial Foundation | | Yoshiyuki Kubota |
| Ministry of Education, Culture, Sports, Science, and Technology (MEXT) | Grant-in-Aid for Scientific Research on Innovative Areas (No. 4103), 24120718 | Yoshiyuki Kubota |
| Ministry of Education, Culture, Sports, Science, and Technology (MEXT) | Grant-in-Aid for Scientific Research on Innovative Areas-Adaptive circuit shift (No. 3603), 26112006 | Yoshiyuki Kubota |
| Ministry of Education, Culture, Sports, Science, and Technology (MEXT) | Grant-in-Aid for Young Scientists (B) 22700321 | Masaki Nomura |
| Ministry of Education, Culture, Sports, Science, and Technology (MEXT) | Grant-in-Aid for Scientific Research on Innovative Areas (No. 3603), 15H01456 | Yasuo Kawaguchi |
| Ministry of Education, Culture, Sports, Science, and Technology (MEXT) | Grant-in-Aid for Scientific Research (A) 25250005 | Yasuo Kawaguchi |

| Funder | Grant reference | Author |
|--------|-----------------|--------|
| Ministry of Education, Culture, Sports, Science, and Technology (MEXT) | Grant-in-Aid for Scientific Research 15K14324 | Yasuo Kawaguchi |

The funders had no role in study design, data collection and interpretation, or the decision to submit the work for publication.

## Author contributions

YK, Conception and design, Acquisition of data, Analysis and interpretation of data, Drafting or revising the article, Contributed unpublished essential data or reagents; SK, Conception and design, Acquisition of data, and Analysis of slice experiments including LM reconstruction and physiological data analysis, Drafting the article; MN, Acquisition of simulation data, Drafting the article; SH, NY, AAM, Acquisition of the EM data and reconstruction from serial EMGs; FK, Analysis of slice physiology data and Neurolucida reconstruction, Drafting the article; JL, Acquisition of the EM data, reconstruction from serial EMGs, Drafting the article; YK, Conception and design

## Author ORCIDs

Yoshiyuki Kubota, http://orcid.org/0000-0002-0950-7460

## Ethics

Animal experimentation: All surgical and animal care methods was performed in strict accordance with the Guidelines for the Use of Animals of IBRO and our institutional Animal Care and Use committee (National Institute for Physiological Sciences) with reference number 14A011. All surgery was performed under ketamine and xylazine, or isoflurane anesthesia, and every effort was made to minimize suffering.

# Additional files

## Major datasets

The following datasets were generated:

| Author(s) | Year | Dataset title | Dataset ID and/or URL | Database, license, and accessibility information |
|-----------|------|---------------|----------------------|------------------------------------------------|
| Kubota | 2015 | CS56-Pyramidal | http://neuromorpho.org/neuroMorpho/index.jsp | Datasets will be available at NeuroMorpho.Org from 28 February 2016. In the meantime the datasets are also available at http://www.nips.ac.jp/circuit/. |
| Kubota | 2015 | CS56-FS-basket | http://neuromorpho.org/neuroMorpho/index.jsp | Datasets will be available at NeuroMorpho.Org from 28 February 2016. In the meantime the datasets are also available at http://www.nips.ac.jp/circuit/. |
| Kubota | 2015 | CS55-Pyramidal | http://neuromorpho.org/neuroMorpho/index.jsp | Datasets will be available at NeuroMorpho.Org from 28 February 2016. In the meantime the datasets are also available at http://www.nips.ac.jp/circuit/. |
| Kubota | 2015 | CS55-FS-basket | http://neuromorpho.org/neuroMorpho/index.jsp | Datasets will be available at NeuroMorpho.Org from 28 February 2016. In the meantime the datasets are also available at http://www.nips.ac.jp/circuit/. |
| Kubota | 2015 | CS56-Pyramidal | https://senselab.med.yale.edu/ModelDB/showModel.cshtml?model=183424 | Publicly available at ModelDB (Accession no: 183424). Also available at http://www.nips.ac.jp/circuit/. |

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
