## [Decision Letter]

Thank you for submitting your work entitled “Functional effects of distinct innervation styles of pyramidal cells by fast spiking cortical interneurons” for peer review at *eLife*. Your submission has been favorably evaluated by a Senior editor and two reviewers, one of whom is a member of our Board of Reviewing Editors.

The reviewers have discussed the reviews with one another and the Reviewing editor has drafted this decision to help you prepare a revised submission.

Summary:

Your manuscript was carefully read by two reviewers who were extremely enthusiastic about the main findings of the study. Both reviewers praised the novelty and importance of the data. Therefore, only few minor comments have been raised which I am attaching below. One reviewer was concerned about the mechanisms which may underlie the decision whether a dendritic or a perisomatic synapse is formed by PV cells. Since distance is an important determinant in this decision process, the reviewer wondered how the PV cell may know how distant its axon is located in relation to the soma. Furthermore, the density of the provided information is very high. The second reviewer was therefore asking for improvement in the presentation of the data to thereby allow a better readability of the manuscript.

*Reviewer #1*:

The manuscript by Kubota et al. describes a high heterogeneity in the size and location of inhibitory synapses provided by fast-spiking interneurons onto target principal cells in the frontal cortex. The relationship between spine location and size of active zones or volume of synapses as well as size of IPSCs is convincing and quite similar to previous investigations on excitatory synapses by [22]. The authors are experts in their field and therefore performed the reconstruction as well as paired recordings and computational analysis on the attenuation and deceleration of IPSCs from the location of their origin to the somatic recording site, quite well. My main concern is related to the question of how does the neuron now whether it should form a perisomatic or dendritic synapses. One possibility could be that certain molecular mechanisms are involved. Alternatively, the overall effect of this form of distance-dependent synapse location on the somato-dendritic domain of target cells may simply depend on postnatal development. During the outgrowth of the axon to it targets, more and more distant dendritic areas are contacted. Although the study is well performed, the explanation for the overall finding is not presented.

*Reviewer #2*:

The study by Kubota and colleagues is of great relevance for the following reasons: 1) It addresses a question regarding the relationship between ultrastructural and functional parameters of defined cortical synapses. At the experimental level the study is based on a combination of EM and electrophysiological measurements that is seldom performed in such depth as in this study. 2) When performed, however, such studies seldom exhibit the excellence of the Kubota et al. study that presents impressing data both quantitatively and qualitatively.

A somewhat comparable study from the Nusser lab had shown previously that release probability is related with the size of the active zone. This investigation focused on glutamatergic terminals. In yet other studies it was demonstrated that the number of glutamatergic receptors in a synapse was correlated with the PSD size. In their study, Kubota and colleagues investigated synapses made by fast-spiking interneurons onto defined layer V cortical pyramidal neurons and correlated structural and functional properties of this synapse.

The inhibitory synaptic strength is progressively smaller from soma to dendritic shaft and spine. The authors report that the scaled inhibitory responses on the three cellular compartments of pyramidal neurons are correlated with the terminal size. The study supports the notion that the findings reveal a general principle according to which the junctional area of inhibitory synapses is correlated to the postsynaptic target size. Furthermore, based on simulations the question is addressed how differences in junctional and synaptic size affect function. The computational data support the notion that functional differences can be accounted for by differences in junctional size. Whether the rule according to which terminal size is related to synaptic efficacy is present also for other types of inhibitoy neurons remains to be seen. At least for fast-spiking interneuron synapses onto pyramidal cells, the authors convincingly show that there is a clear relation between size and efficacy of synaptic terminals.

The experimental part of the study is of excellent quality as one is used from previous studies from these authors. The readability of the manuscript is sometimes difficult but this is not a criticism as it is in great part due to the wealth of information. The authors might nevertheless try wherever possible to present the data such that it is easier for the reader to follow. For instance, the order of the panels in Figure 2 needs to be rearranged.

In summary, studies like this one are direly needed if we want to eventually determine general rules based on which distinct operations can be deduced from the underlying synaptic structures. This knowledge is a prerequisite for properly linking the contribution of defined synapse activity with the next complex level of function, namely that of neuronal microcircuits. I highly recommend the acceptance of this manuscript in the submitted version.

---

## [Author Response]

Reviewer #1:

*The manuscript by Kubota et al. describes a high heterogeneity in the size and location of inhibitory synapses provided by fast-spiking interneurons onto target principal cells in the frontal cortex. The relationship between spine location and size of active zones or volume of synapses as well as size of IPSCs is convincing and quite similar to previous investigations on excitatory synapses by*
[22]*. The authors are experts in their field and therefore performed the reconstruction as well as paired recordings and computational analysis on the attenuation and deceleration of IPSCs from the location of their origin to the somatic recording site, quite well. My main concern is related to the question of how does the neuron now whether it should form a perisomatic or dendritic synapses. One possibility could be that certain molecular mechanisms are involved. Alternatively, the overall effect of this form of distance-dependent synapse location on the somato-dendritic domain of target cells may simply depend on postnatal development. During the outgrowth of the axon to it targets, more and more distant dendritic areas are contacted. Although the study is well performed, the explanation for the overall finding is not presented*.

The reviewer raised an interesting point that we followed up on. We added one paragraph in Discussion to state how can our results contribute to this very important issue.

Reviewer #2:

*The experimental part of the study is of excellent quality as one is used from previous studies from these authors. The readability of the manuscript is sometimes difficult but this is not a criticism as it is in great part due to the wealth of information. The authors might nevertheless try wherever possible to present the data such that it is easier for the reader to follow. For instance, the order of the panels in*
Figure 2
*needs to be rearranged*.

We apologize for the confusing arrangement of the panels in the Figures in the original manuscript. We rearranged the panels in Figure 1 and Figure 2 in order for the reader’s easier understanding.